# Reconstructing archaeological palaeolandscapes using geophysical and geomatic survey techniques: An example from Red Lily Lagoon, Arnhem Land, Australia

**Jarrad Kowlessar**[1], **Ian Moffat**[1]*, **Daryl Wesley**[1], **Mark Willis**[1,2],
**Shay Wrigglesworth**[3,4], **Tristen Jones**[5], **Alfred Nayinggul**[3], **the Njanjma Rangers**[3]

**1** Archaeology, College of Humanities, Arts and Social Sciences, Flinders University, Bedford Park, South Australia, Australia, **2** Sacred Sites Research, Albuquerque, New Mexico, United States of America, **3** Njanjma Rangers, Gunbalanya, Northern Territory, Australia, **4** Kakadu National Park, Jabiru, Northern Territory, Australia, **5** Department of Archaeology, The University of Sydney, Camperdown, New South Wales, Australia

☯ These authors contributed equally to this work.

* ian.moffat@flinders.edu.au

## Abstract

Arnhem Land is a key region for understanding the Pleistocene colonisation of Australia, due to the presence of the oldest sites in the continent. Despite this, conventional archaeological survey has not been effective at locating additional pre-Holocene sites in the region due to a complex distribution of geomorphic units caused by sea level rise and coastal aggradation. This research uses geophysical and geomatic techniques to map the subsurface distribution of the geomorphic units in the Red Lily Lagoon region in eastern Arnhem Land. This reveals a complex Pleistocene landscape, which offers the potential to locate additional archaeological sites and so reveal more about the lifeways of the earliest Australians.

## Introduction

Archaeological attention is often focused on occupation sites, yet in context of palaeolandscapes that are no longer visible in the surface morphology, archaeological understanding of these records is significantly limited. This is especially challenging in the long-lived cultural landscapes of Australia's north coast, which have undergone significant palaeogeographic change during human occupation.

The detailed reconstruction of past landscapes facilitates effective predictive modelling of archaeological site location and allows archaeological sites to be placed in their physiographic and environmental context. Environmental context provides an important means to interpret the presence and provenance of cultural material.

The Greater Red Lily Lagoon Area (GRLLA) (Fig 1) is a province of exceptional archaeological significance in Arnhem Land, northern Australia. It is situated at one of the easternmost

**Data Availability Statement:** All data are held in the Open Science Framework public repository (https://doi.org/10.17605/OSF.IO/8PJER).

**Funding:** JK is the recipient of a Flinders University Postgraduate Scholarship (flinders.edu.au). IM is the recipient of an Australian Research Council Discovery Early Career award (project number DE160100703, arc.gov.au) funded by the Australian Government, a George Chaloupka Fellowship from the Museum and Art Gallery of the Northern Territory (magnet.net.au) and a Flinders University Early Career Researchers Award (flinders.edu.au). DW is the recipient of an Australian Research Council Discovery Early Career award (project number DE170101447, arc.gov.au) funded by the Australian Government and a George Chaloupka Fellowship from the Museum and Art Gallery of the Northern Territory (magnet.net.au). TJ is the recipient of a George Chaloupka Fellowship from the Museum and Art Gallery of the Northern Territory (magnet.net.au). The funders had no role in study design, data collection and analysis, decision to publish, or preparation of the manuscript.

**Competing interests:** The authors have declared that no competing interests exist.

extents of the East Alligator River floodplain, where the modern river and abandoned palaeo-channels are adjacent to the sandstone escarpment of the Arnhem Plateau geological formation. This significant boundary between the low laying floodplains and the sandstone highlands of the Arnhem Plateau have been occupied by humans for over 60,000 years [1] and is the site of countless significant archaeological sites including some of the most iconic rock art panels in Australia [2–14]. This is a key landscape for understanding the early human occupation of Australia, with four sites greater than 20 ka having been found within 15 km of the GRLLA [1,15,16 (p 102), 17 (pp 75, 110–45)].

Interpretation of the art and material culture of Indigenous people in this area can be enhanced by developing a better understanding of the past landscapes and their relationship to the locations of human activity. The people living in the GRLLA have seen a significant environmental boundary affected by the climate changes of the last glacial cycle. Over the period of human occupation in this region, the landscape has been subject to dramatic environmental change, transitioning from a semi-arid location more than 350 km from the coast during Marine Isotope Stage 3 (MIS 3) to becoming a coastal mangrove swamp at the Holocene sea level highstand and then a seasonally inundated floodplain more than 40 km from the coast today [16,18,19]. These environmental changes are broadly mirrored in the subjects and chronology of the rock art in the area, presenting an Indigenous vision of this landscape and its environmental changes [2]. Similarly, excavated cultural materials from occupation sites in the East Alligator River area have provided important insights about human activity over the last 65,000 years and many behavioural adaptations to these changing environments over this time [1,13,15,16,17,20–25].

A key interest for researchers in Arnhem Land is locating additional Pleistocene cultural deposits, such as those located at Madjedbebe [1]. While many of the early excavations in the region located sites with pre-Last Glacial Maximum (LGM) ages [16,17,20,26], subsequent research has mainly yielded sites with Holocene chronologies [13,23,24]. While sites of any age provide important information about the rich human history of the region, the location of new sites of Pleistocene age will afford new sequences to compare to previous excavation results (particularly in terms of chronology) [27–31] as well as providing further information about pre-Holocene life histories. Further, all Pleistocene sites excavated in this region have been rockshelters and so provide a selective understanding of the life history of the people who occupied this landscape. We contend that the inability of recent excavations to replicate the antiquity of Madjebebe is based on the geomorphology of the excavation locations chosen rather than the presence of suitable sites in the region, which can be addressed by developing a more detailed model for landscape evolution.

Geophysical techniques provide a non-invasive, time- and cost-effective means of investigating the subsurface [32,33] and are widely used for archaeological and geological investigations [34–39]. Electrical Resistivity Tomography (ERT) is a geophysical technique that measures the resistance of a volume of the subsurface by transmitting electrical current between electrodes [32,33 (pp 208–12)]. The depth of each measurement is controlled by the electrode spacing, and multiple measurements with different electrode spacings are combined to construct a 2D profile or 3D data volume. A mathematical modelling procedure called inversion is used to estimate the conditions that best reflect the measurements obtained, and so produce a profile of resistivity that can be interpreted for geological features which can inform models of palaeolandscapes [40].

Modern geomatic approaches allow geophysical data to be combined with detailed surface, and elevation modelling. Combining surface and subsurface models allows a detailed interpretation of palaeogeographic changes and even the estimation of past land surface models creating detailed models of palaeolandscapes.

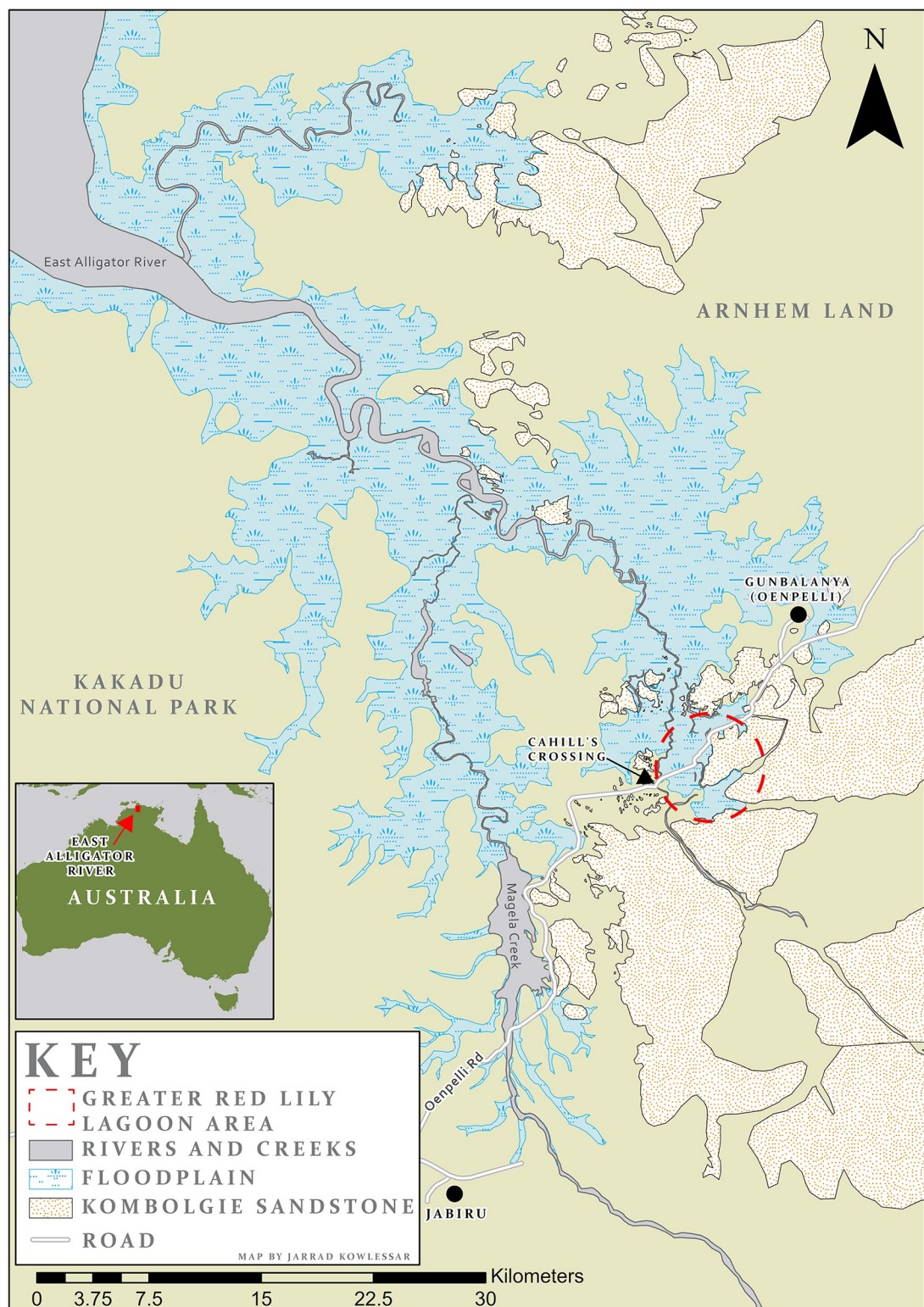

**Fig 1. East Alligator River floodplain and the Arnhem Plateau.**

Whilst the broad environmental history of the region over the period from the LGM to today has been characterised for the South Alligator River and Magela Creek floodplains [19,41,42], the subsurface geomorphology of the floodplain east of the East Alligator River is yet to be investigated. In order to address this research gap, mapping of the sedimentary facies and bedrock geomorphology of the GRLLA using the non-invasive geophysical method of Electrical Resistivity Tomography (ERT) has been undertaken. The results were used to analyse the distribution of rock art sites with reference to the palaeolandscape of the region and to model where new archaeological sites of Pleistocene age may be located. This work showcases the importance of accurate models of past landscapes through a detailed study of the culturally important Red Lily Lagoon section of the East Alligator River floodplain.

## Environmental background

**North Australian climate history.** Northern Australia has experienced wet-dry seasonality and monsoon events since before 150 ka, the seasonality and severity of these events were affected by global climate patterns [43]. During MIS 3 and before the transition into the LGM, the region had a humid but cooler climate than present day [44–47]. This period saw high fluvial activity across the continent [44,48,49]. This has been demonstrated for areas around East Alligator River with palaeochannels dating to this time showing multiple cycles of incision and infilling [50].

During MIS 3, Australia was wettest between 49 and 40 ka [51] before becoming increasingly dry leading up to and during MIS 2 (29–14 ka) [47,52]. The increasing dryness was also reflected in monsoon events that were less severe than those of the modern day [47,52].

This LGM period saw dramatic changes to the global sea level and temperature as ice caps increased to their maximum extent. This had profound effects on the climate of northern Australia. During the LGM, sea level was at least 120 m lower in northern Australia [53], which exposed large areas of the continental shelf and opened land bridges on the Sahul Shelf between New Guinea and Australia. Exposing this land bridge resulted in major changes to the ocean circulation that greatly restricted the warm shallow waters that feed the tropical rainfall in the region [44,52]. This caused a drier period in Northern Australia. The dramatically different base level also had significant impacts on the geomorphology of fluvial systems in the region.

MIS 2 contrasts with the wet tropical climate that preceded in MIS 3 as well as the contemporary tropical climate which followed during MIS 1. It was terminated by a warming event that coincided with sea level rise and flooding of the Sunda shelf [44,52]. This flooding of the Sunda Shelf along with the deglacial warming in the region led to the return of the seasonal monsoon which characterises the contemporary climate of northern Australia [52].

**West Arnhem Land geology and geomorphology.** The GRLLA study area is located at the boundary between the Archean to Paleoproterozoic Pine Creek Orogen and the Palaeo-Mesoproterozoic McArthur Basin geological provinces, which correspond with the distribution of the Arnhem Land floodplains and plateau (respectively). The Paleoproterozoic Kombolgie Formation dominates outcrops in the Arnhem Land Plateau (see Fig 7). This formation is made up of fluvial sandstones as well as some extrusive volcanic units. The sandstones of the Kombolgie Formation are a ferruginous fine- to course-grained quartz arenite. There are four distinct sandstone lithologies within the Kombolgie Formation: Marlgowa, Gumarrirnbang, Mamadawerre and McKay Sandstones which are differentiated principally by quartz grain size [54–56].

The Kombolgie Formation of the McArthur Basin unconformably overlies the Pine Creek Orogen units, which extend westward. The Pine Creek Orogen is made up of sedimentary and

volcanic rocks which have undergone metamorphism and deformation and then deep weathering through laterization. This laterized surface is known as the Koolpinyah surface, which forms broad undulating lowlands that reach the coast and continue as far west as Darwin (Fig 1). The Koolpinyah surface underlies the coastal alluvium of the floodplains of the Alligator rivers. Discrete outcrops of Archean basement inliers are also present locally in the floodplain [54–57].

The denudation rate of the Koolpinyah surface has been shown to be up to eight times higher than that of the Kombolgie Formation sandstone of the Arnhem Plateau. Denudation rates of these two surfaces have been shown to be related to the climate events of the glacial cycles of the past 500 ky [55]. The two major drivers of denudation are cycles of valley incision during sea level lowstand and valley infilling during highstands, along with increased fluvial activity associated with dry/wet climate transitions of the glacial cycles [55 (pp 886–7)]. With its much lower denudation rates, the Kombolgie sandstone forms a barrier to the marine incursions that have occurred repeatedly over the Quaternary period. The difference in denudation rates between the two land surfaces increases their relief relative to one another [55]. The floodplains of the Alligator rivers are formed in the low-lying valleys of the Koolpinyah surface. These lowlands contain meandering, tidally impacted fluvial systems that have accumulated sediments eroded from the Kombolgie and Koolpinyah land surfaces. In this way, the low-lying valleys of the floodplains act in contrast to the eroding uplands of the Koolpinyah surface and are gradually increasing in elevation as sediments collect there.

At the edges of the sandstone escarpments, colluvial sands created by the eroding sandstone collect as slope aprons at the foot of the escarpments and as alluvial fans that fill valleys throughout the Arnhem Plateau [55]. Some of these sediments are mobile enough to reach the floodplains. The sand component of floodplain sediments comes from the erosion of the Kombolgie Formation whilst silt, clay and terrigenous solutes come from the erosion of the Koolpinyah surface [41,42,55]. Fig 2 demonstrates the relationships between the Kombolgie sandstone escarpment, the Koolpinyah surface and the floodplains of the East Alligator River.

The lower-lying areas of the Koolpinyah surface, which are currently floodplains of the Alligator rivers, followed similar wide flat valley forms before the sea level rise at the end of the LGM but were less elevated [41,42,50,58]. These alluvial valleys are characterised by unconsolidated sediment unconformably overlaying bedrock [59 (p 1141)], in this case the Pine Creek Orogen conglomerate capped by the laterized Koolpinyah surface [41,42,58].

The sedimentary record of Megela creek demonstrates that the Alligator rivers region was a fluvial landscape for the entirety of the Quaternary period [50]. Fluvial activity has been heavily influenced by the glacial cycles over this period. The Quaternary has demonstrated cycles of channel incision (during glacial low stands) and infilling (during deglacial transgressions). Fluvial activity has also impacted by the climatic (changes in precipitation) and eustatic (and associated base level) changes of the glacial cycles. This process has left the valley weakly incised in the Koolpinyah surface, as a result of channel incision during periods of lowered base level [50]. Older sediments remain in terraces along the valley edges but the sediments in the central parts of the valleys are generally reworked by recent fluvial activity. The most recent palaeochannels were formed during the last glacial cycle.

The sea level transgression following the LGM flooded the valleys below the Arnhem Land escarpment. This flooding occurred between 8 and 6 ka in the South Alligator River valley and is thought to be similarly timed in the East Alligator River valley [42(p 261), 50, 58(p 19)]. The transgressive flooding first reached Magela Creek around 7.7 ka and continued until between 5.5 and 7 ka [41(p 90)].

**Alligator rivers morphology.** The contemporary South and East Alligator rivers are typical tide-dominated estuaries with a funnel-shaped geometry where they enter the ocean and

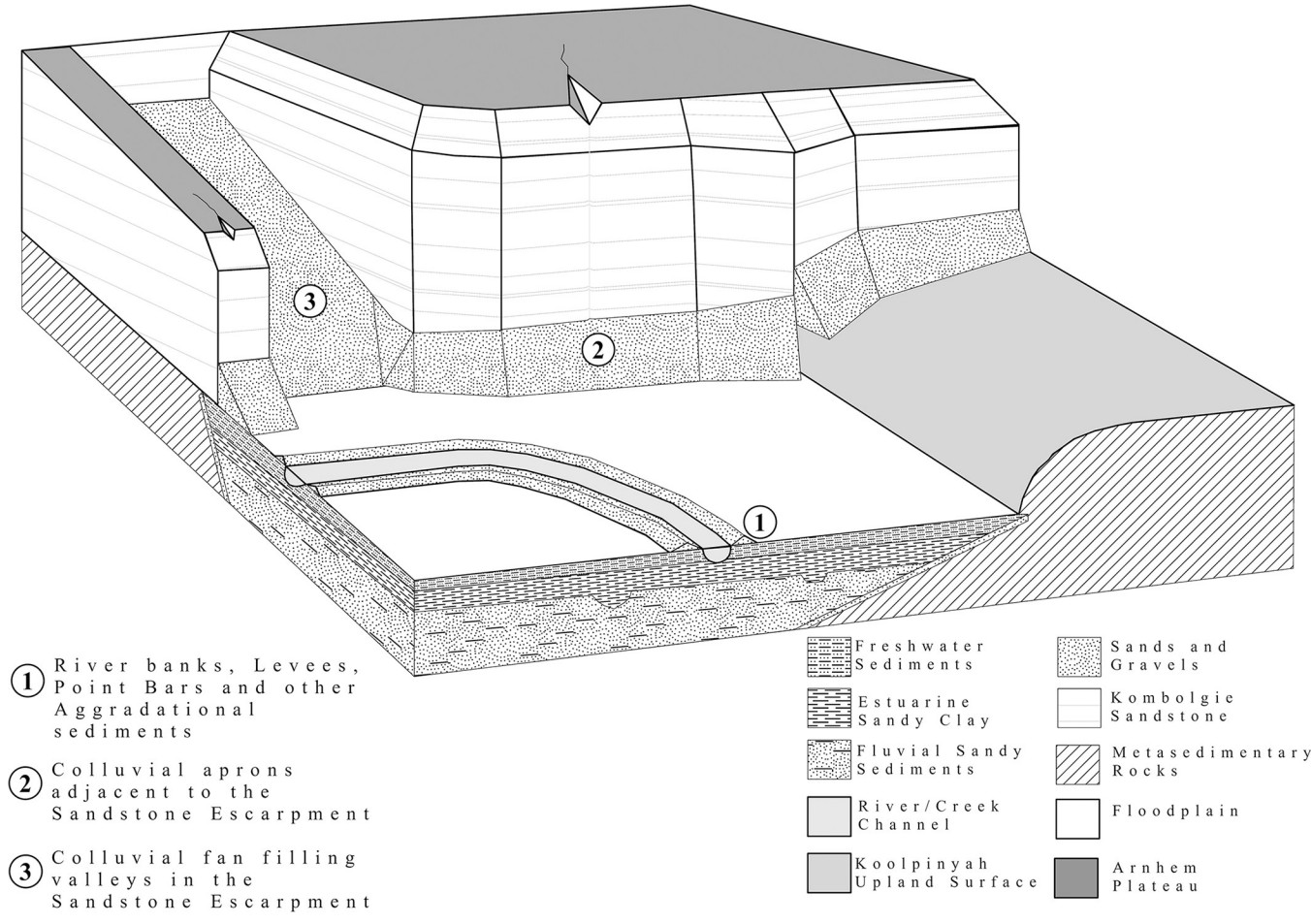

**① River banks, Levees, Point Bars and other Aggradational sediments**

**② Colluvial aprons adjacent to the Sandstone Escarpment**

**③ Colluvial fan filling valleys in the Sandstone Escarpment**

Freshwater Sediments

Estuarine Sandy Clay

Fluvial Sandy Sediments

River/Creek Channel

Koolpinyah Upland Surface

Sands and Gravels

Kombolgie Sandstone

Metasedimentary Rocks

Floodplain

Arnhem Plateau

**Fig 2. Key geomorphological features of the East Alligator River floodplain and Arnhem Plateau.**

alternating straight and meandering sections moving landward [59]. This morphology is a result of interacting marine and fluvial forces within the estuary [59]. The tidal limit of the East Alligator River estuary occurs a short distance into the sandstone plateau, where elevation increases from the flat floodplain valley floor [60]. Fluvial energy (river currents) decreases seaward. In the middle section of a tide-dominated estuary, fluvial and tidal currents meet with roughly equal energy, and the river tends to meander. The mixed-energy zone is the lowest-energy zone in the estuary and therefore is the site of the deposition of finest-grained sediments from both marine and fluvial sediment loads, with grain size increasing both seawards and landwards from this central section [59(pp 1134–6)]. The sinuosity (meandering form) of the river in these low-energy zones is thought to be caused by frictional energy gradients forming point bars on small curvatures in the channel path. The gradient of depth across the channel creates a pressure gradient in the current which curves the water with increasing force around the bends of these point bars cutting into the outside shoulder of the channel. Sinuosity will develop into these distinct meandering sections where the mixed-energy zone of the tide-dominated estuary occurs in a section of the river valley where the channel is unconfined [59(p 1136)].

The landward straight section which is upstream of this mixed-energy zone is dominated by fluvial river currents, and the seaward straight section is dominated by marine tidal currents

[58(pp 28–32), 59(pp 1134–6)].The straight-meandering-straight, tidally dominated estuary morphology can only develop with a relatively stable sea level as the mixed-energy zone needs to be stable in location for some time before the physical forces (tidal and fluvial currents) can form this morphology. During the transgressive phase of sea level change, this zone will move rapidly and cause less morphological changes as estuaries are formed from the existing fluvial system and land surface configuration [59(p 1140),61]. The morphological changes brought about by the three distinct energy zones associated with tide-dominated estuaries will occur once sea level is stable or once the rate of sediment supply exceeds the rate of relative sea level change and infilling of the estuaries occurs [59(p 1140),61]. The three energy zones will gradually migrate seaward as the coastline (and zone of tidal influence) progrades.

This estuary morphology has been further characterised in the case of the much more intensively studied South Alligator River, providing an analogue of the morphology of the East Alligator River [60,61(p 755)]. Along the South Alligator River, the following channel forms have been identified, listed in order moving landward: estuarine funnel (river mouth and deltaic shelf), sinuous meandering segment (mixed-energy zone), cuspate meandering segment and the upstream tidal channel [58,61(pp 740–1)]. Fig 7 demonstrates these channel forms in the contemporary East Alligator River.

The cuspate meandering and upstream sections of the tidal channel together form the fluvially dominated landward straight section of the South Alligator River estuary [58(pp 43–5),59]. The cuspate meanders have the form of pointed inside river bends with wide reaches. Mid-channel shoals are usually present between these pointed cusps. These sections form in areas of past meanders where the mixed-energy zone has moved seaward due to progradation and now the channel is fluvially dominated. Increasing the energy and reducing the sediment deposition in the meandering zone causes channel bank erosion following the sharp pressure gradient of the original meander's point bar, widening the channel at this bend and cutting the point bar to create the mid channel shoal [59,61(pp 741,55)]. The upstream tidal channel describes the segment of the estuary with the greatest fluvial influence, which extends from the cuspate section to the tidal limit. This section has meanders with sharp bends and long straight reaches [58,61(p 741)].

Driven by sea level change following the LGM, the South Alligator River region went through a developmental sequence, described by four major phases: Transgressive phase, Big Swamp phase, Sinuous phase and Cuspate phase.

The Transgressive phase is described as the period of post glacial sea level rise which transgressed the contemporary areas of the South Alligator River between 8 and 6.8 ka [61(p 735)]. A number of models for this transgression have been put forward with the favoured model describing a rapid transgressive flooding occurring over almost the entire extent of the modern floodplains [61(pp 117–9)]. This model has been supported by the presence of mangrove materials in palaeochannels of the South Alligator River dating to between 6.8 and 5.3 ka. Sea level rise ceased around between 6.5 and 6 ka in this area when present day sea level was reached [58(p 127)].

The Big Swamp phase is described as a period following marine transgression and the establishment of estuaries by 6.8 ka, when mangroves flourished across the South Alligator River region [19,42,58,61,62]. Mangrove sediments dating to this period have been uncovered across nearly the entire floodplains of the South Alligator River. With sea level stabilisation and the sedimentation of estuaries increasing due to mangrove development, the movement of marine water began to be restricted and reduced, and the mangrove habitat receded [61]. Whilst the development of mangroves has been suggested to have been a rapid and far-reaching process [58] the decline of the mangroves has been characterised as more complex and locally variable

across the Alligator rivers [25,41]. The sedimentation of individual channels may have produced poor drainage and hypersaline mudflats which cause rapid localised mangrove decline.

The Sinuous phase is described as a period of river channel migration through the far-reaching mangrove sediments deposited through the Big Swamp phase. This phase has left remnants of past meanders as oxbow lakes along the length of the South Alligator river showing the locations of past mixed energy zones throughout the history of progradation.

The Cuspate phase is described for its distinct channel morphology of eroded riverbanks, and sharp pointed cusps in river bends. As the energy zones prograded seawards, the previous low energy meander sections began to enter the higher fluvial energy zone and bank erosion morphed the channel in these sections into cuspate forms beginning around 2.5 ka [59,61(pp 735,41,55)].

The morphology of the plains in the river valley lowland of the South Alligator River region also changed over these distinct phases. During the LGM these areas were low laying valleys with fluvial sediments overlaying the bedrock. After the Transgressive phase, these valleys became deltaic-estuarine plains. Once these plains filled with mangroves during the Big Swamp phase sediment accretion levelled these regions into flat topography. During the Sinuous phase as mangroves declined, these regions became saline mudflats. The freshwater dominance of the monsoon climate especially during the cuspate phase caused seasonal flooding of the mudflats transforming them to the contemporary freshwater floodplains. These freshwater floodplains are populated by grasslands as well as open melaleuca paperbark forests and areas of semi-permanent standing water near the valley edges termed 'back water swamps' [41,61].

**Sedimentary facies and stratigraphy.** The stratigraphy of the floodplains of the South Alligator River and Magela Creek have been well characterised through coring, radiocarbon dating and pollen analysis [41,42]. This coring was used to characterise the Transgressive, Big Swamp, Sinuous and Cuspate phases of the South Alligator River (and East Alligator River by extension). Table 1 shows the identified layers and their ages, depths and elevation, and Fig 3 shows the full elevation ranges observed in the Magela floodplain for each surface and denotes the range of the top of each layer's surface. Fig 2 shows the major depositional layers in their landscape context relative to the Kombolgie escarpment and the eroding upland Koolpinyah surface.

*Bedrock and basal sediments.* Wasson [41] identified the bedrock depth through the Magela Creek floodplain as between 7 m (~-1 m elevation Australian Height Datum [AHD]) and 29.3 m (~-24 m elevation AHD). Bedrock was found to increase in depth below the floodplain near the modern East Alligator River, with its deepest measured depth being on the north side of the river in the centre area of the floodplain valley.

Above the bedrock, a layer of basal sands and gravels with an approximate thickness of 4 m was identified. At this layer's shallowest point, the top surface of this strata was sampled at a depth of 5 m (~0.5 m elevation AHD) [41(p 124)]. At its deepest point sampled in the Magela floodplain, the top surface of this layer was found at a depth of 27.4 m (~-22.4 m elevation

**Table 1. Major depositional layers age, depth from surface and elevation of the top surface level of each of these layers as observed throughout the Magela Creek [41].**

| Depositional layers | Age | Depth to the top of the layer surface | Elevation (AHD) of the top of the layer surface |
|---|---|---|---|
| Bedrock | 2020 to 1800 Ma | 7 to 29.3 m | ~-1 to -24 m |
| Basal sands and gravels | Unknown but assumed to be pre-Holocene | 5 to 27.4 m | ~0.5 to -22.4 m |
| Pre-transgression land surface | >7.7 ka | 2.5 to 11 m | 3 to -8 m |
| Mangrove muds and clay | 7.7 ka | ~1.5 m | 1.5 to 3.8 m |
| Freshwater floodplain clay | 5 to1.7 ka | 0 m | ~3.5 to 5.5 m |

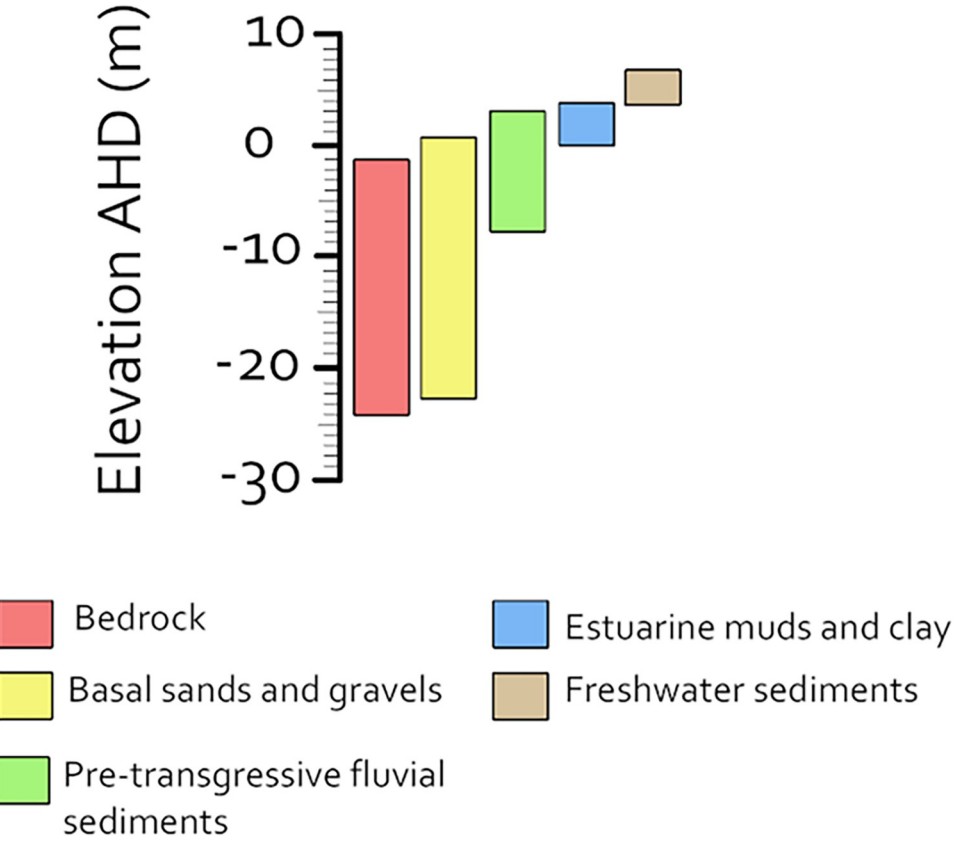

**Fig 3. Elevation ranges as reported by Wasson [41] for each depositional layer of the Magela Creek floodplain.**

AHD) and the bottom of the layer was at a depth of 29.3 m laying unconformably on the bedrock [41(p 28)].

The basal sediment layer identified above the bedrock in the floodplains of the South Alligator River is comprised of gravels, quartz sands and weathered clays; these sediments are interpreted as pre-Holocene surfaces [42,58,61(p 743), 62] but no dating has been undertaken on them.

*Freshwater fluvial and colluvial sediments.* Above the basal sands and gravels in Magela Creek is a layer of sands and clays that have been characterised as fluvial sediments deposited in a system with no marine influence [41(p 29)]. These sediments are interpreted to represent the land surface prior to the marine transgression and therefore have a Pleistocene to early Holocene minimum age [41(p 90)]. These sediments formed fluvial terraces that were buried by younger sediments [41(p 29)]. The top surface of this layer of pre-transgression fluvial sands and clay was found at a depth of 2.5 m (3 m elevation AHD) at the shallowest point sampled [41(p 124)] and a depth of 11 m (-8 m elevation AHD) at the deepest point sampled [41(p 135)].

In the South Alligator River, the basal sediments include layers of clean white sands which are interpreted as early Holocene or Pleistocene alluvial or colluvial deposits [42,58,61(p 743),62]. The principal control on the distribution of the sediments appears to be the topography of the bedrock surface [41(p 29)].

*Mangrove mud.* Overlying the fluvial sediments of the floodplains of South Alligator River and Magela Creek is a layer of 'blue-grey' soft clay mud [41(pp 55–6,141),61]. Palaeobotanic

analysis of samples from both areas have confirmed this to be mangrove mud, rich with in situ mangrove stumps, macrofossils and pollen [41,61]. Within the South Alligator River region this layer has a base level between -12 and -10 m AHD. Along Magela Creek, the bottom of this layer was found at an elevation of -10 m AHD (at around 15 m depth) at its deepest and found all the way to 3.8 m AHD (approximately 8.5 m depth) further inland in the upstream end of the Magela floodplain [41]. This layer has a thickness which varies between 4 and 14 m along Magela Creek (confined by the gradient of the bedrock depth) [41] and between 2 and 8 m in the South Alligator River region [61].

The Transgressive phase occurred between 8 and 6.8 ka with the Big Swamp phase lasting from 6.8 to 5.3 ka [42]. The transgression first reached Magela Creek around 7.7 ka as demonstrated by dated mangrove pollen at the base of the blue grey clay layer where the creek first branches from the East Alligator River [41(p 90)]. Mangroves reached the upstream extents of Magela Creek by 4.4 ka [41(pp 55–6,141)]. The contemporary elevation extent of mangrove distribution is between 3.7 and -1.0 m from mean sea level [58]. This ecological range appears to have been consistent in the region since at least the LGM [41(p 90)].

*Laminated channel sediments*. Laminated lenses of bluish-grey silt and fine-grained sand have been identified in the subsurface adjacent to the modern South Alligator River channel. These have been argued to represent lateral accretion deposits associated with channel migration and associated outbuilding sediments on river point bars [61(p 743)]. These sediments have been radiocarbon dated to between 5.1 and 2.9 ka in sections occurring between palaeo-channels and the contemporary river channel.

*Transitional sediments*. The blue-grey clay is overlain by a considerably thinner layer of grey clay. This layer was argued to be related to a transition from the Big Swamp estuarine environment to the modern freshwater dominated seasonal wetland conditions [41(pp 29–30)]. This layer was not identified directly throughout the South Alligator River region. However, a layer of undifferentiated sediments was identified occurring at comparable depths and stratigraphic sequence (overlying blue-grey mangrove mud) [61(p 743)].

*Floodplain sediments*. The surface layer of the Magela floodplain was described as 'dark brown/black clay'. Rich in organic material, this layer does not display visible sedimentary structures and is likely to be formed by the deposition of floodplain sediments and the in situ breakdown of grasses and sedges [41(p 31)]. This floodplain clay layer is approximately 0.5–1.5 m thick but is difficult to distinguish from underlying transitional sediments [61 (p 743)].

## Archaeology background

**Archaeology of the East Alligator River catchment.**   Nine archaeological sites have been excavated around the estuarine portion of the East Alligator River catchment [1,13,15–17,20–25]. These sites account for the record of material culture of this area and are dominantly characterised by faunal remains (including shell and bone), stone artefacts and charcoal [1,13,15,17,20–25,63]. The locations of each shelter can be seen on Fig 1. All of these sites are sandstone rockshelters on the margins of the Kombolgie formation that overlook the East Alligator River or Magela Creek floodplains. These shelters are situated at different elevations, but all are located on colluvial aprons surrounding the sandstone margins. Table 2 describes each site's onset of occupation (based on dated material). These excavations, whilst limited in number, have significantly contributed to the discussion of human occupation of northern Australia over the last 65,000 years. This includes the late Pleistocene through to the modern period with particular detail for the mid-late Holocene following the marine transgression. This section will briefly review these material records and surrounding accounts of human activity.

**Table 2. Excavations and derived occupation dates throughout the East Alligator floodplains region (Previously uncalibrated dates for Malangangerr, Nawamoyn and Paribari were calibrated using rCarbon with the SHCal20 calibration curve and reported with 1 sigma ranges).**

| Site | Source(s) | Maximum occupation |
|---|---|---|
| Madjedbebe (Malakununja II) | [1,15,20,21,26] | 65.0 ± (3.7, 5.7) kyr. |
| Ngarradj Warde Djobkeng | [16(p 102), 26(pp 29–36,64–6)] | 26,000 cal. BP |
| Malangangerr | [17(p 75)] | 30,708–30,594 cal. BP |
| Nawamoyn | [17(pp 110–45)] | 26,034–25,256 cal. BP |
| Bindjarran | [23(p 108)] | 13,140–12,771 cal. BP |
| Birriwulk | [24] | 5290–4970 cal. BP |
| Paribari | [17(pp 45–74)] | 3,447–3,215 cal. BP |
| Ingaanjalwurr | [22] | 1,900–1,300 cal. BP |
| MN05 (Red Lily) | [13] | 795–950 cal. BP |

*Pre-transgression (65,000 to 9000 years).* Madjedbebe (previously known as Malakununja II) has produced the earliest dated occupation for the region and Australia [1]. The archaeology of Madjedbebe is characterised by an extensive stone artefact assemblage, faunal remains and shell middens. The site is situated in a sandstone rockshelter and sits on a colluvial apron at approximately 20 m elevation (AHD) adjacent to the Magela Creek floodplain [1(p 306)] at the northern edge of the sandstone escarpment. The site is elevated compared to the floodplain, which is ~3–4 m elevation (AHD). Aboriginal occupation at Madjedbebe has been dated from 65 ka and shows evidence of changing lithic technologies as well as ochre collection and grinding/processing of resources during this early time [1,64–66]. Organic material preservation is largely limited to the Holocene. However, organic materials preserved from the early Transgressive phase can be used to infer some of the land use and subsistence practises during the early Holocene portion of the pre-Transgressive phase (see following section). Occupation during the pre-transgression has been recorded in Ngarradj Warde Djobkeng from 26 ka [16 (p 102), 26(pp 29–36,64–66)], Malangangerr from 24.8 ka [17(p 75)], Nawamoyn from 21.4 ka, [17(pp 110–45)], and Bindjarran from 13–12 ka [23(p 108)].

Plant remains recovered from sediments of 65,000–53,000 years ago from Madjedbebe suggest an open forest and woodland and/or monsoon vine forest environment [67].

*Transgressive phase (8.5–8 ka to 6.8 ka).* The Transgressive phase's period of occupation at Madjedbebe has abundant vertebrate remains demonstrating that small marsupials from grasslands, woodlands, and dry eucalypt woodland habitats were the primary food source during this period [25(pp 197, 49–50)]. At the beginning of the transgression both estuarine and freshwater molluscan resources were a secondary source of food but as the mangroves became established towards the end of the Transgressive phase, resource harvesting of estuarine molluscs became dominant [25(p 198)].

The earliest known material record of human interaction with the emerging East Alligator River environments of the Transgressive phase comes from dated shell midden contexts at Madjedbebe and Nawamoyn [17(p 118), 25(pp 115,95)]. The shell assemblage from Nawamoyn broadly agrees with those from Madjedbebe but was less detailed in its recording and subsequent study [17,25]. The midden formation at both sites coincides with the earliest mangrove development for the East Alligator River valley around 7600 cal BP [17(p 118),25,41]. The midden materials dated to the Transgressive phase show the presence of both freshwater and estuarine molluscs. Estuarine molluscs were more dominant within the midden and increased in exploitation over the Transgressive phase. Analysis of the shell of this period

indicates that Madjedbebe was only used ephemerally during this time. Mollusc species present indicate that all areas of the developing mangrove forests were foraged during this phase whereas later periods saw only specific zones of the mangrove forest foraged [25(p 196)]. Woo [2] suggested that this broad mangrove foraging strategy was only possible owing to the early developmental stage of the mangrove habitats with a more open forest structure allowing greater accessibility. As these mangrove forests are established the seaward zones of these forests became increasingly dense, thereby restricting access to the unique fauna of those zones [25(pp 196–7),68].

Florin et al. [69] has investigated palaeoclimate for this region using the novel analysis of Pandanus nutshell (*Pandanus spiralis*) remains excavated from Madjedbebe. Variations within the $\delta^{13}$C within this dated assemblage were used as a proxy for mean annual precipitation. These data suggested a period of increased precipitation between 9.7 and 7.1 ka which then steadily declined towards the modern day. Florin et al. [69] demonstrated the reliability of this approach using modern Pandanus nut shells collected across a transect with a significant precipitation gradient. The modern analysis demonstrated accurate predictions of precipitation in areas with well-drained soils. However, precipitation was overestimated in areas with standing water on the surface. Given the coincidental timing of the predicted increase in precipitation with the marine transgression in this region, the variation in $\delta^{13}$C at Madjedbebe may be better explained by a change to the local hydrology related to the short distance from the site to the fresh/saltwater interface.

Additionally, analysis of grinding stones from Madjedbebe has shown that waterlily (*Nymphaea violacea*) was being processed at the site at 8320 cal BP [70]. This plant species grows in low-energy freshwater environments and was probably available locally in the East Alligator River due to an increase sinuosity in this system due to the rapid increase in fluvial base level associated with the transgression.

*Big Swamp phase (6.8 to 4–5 ka).* The Big Swamp phase at Madjedbebe coincides with a shift to estuarine mollusc foraging as the dominant food source, particularly those from the landward fringes of the mangrove forest [16,25,68]. The development of these extensive forests had considerable impacts on mobility, particularly in terms of restricting access to the seaward portions of mangrove habitat. Whilst these shifts in mangrove zone exploitation seem to be present in all excavated middens in the area, the timing of these shifts differ between sites [25(p 228)]. This is indicative that mangroves forest structure development was varied throughout the region, likely representing specific landscape transitions and subsequent forest development [25(p 228)].

Foraging strategies were highly specialised and focused on molluscs by the peak of the mangrove development [25(p 228)]. Shell tools and scrapers appear in the site assemblages. This coincides with an observed decrease in lithic production in the area during this period [17(pp 250–1), 25(p 225)]. Imported mollusc species have been observed among the assemblages at Ngarradj Warde Djobkeng, Nawamoyn, and Malangangerr. These species are all of coastal marine origin and therefore transported or traded to this site over significant distances (between 50–300 km) [25(p 225)].

This period is also coincident with the appearance of bifacial points in many of the rockshelters in the region around 5–7 ka [68(p 98)]. It is unclear whether this occurrence is best timed with the Big Swamp or its subsequent decline but is likely to represent technology changes in the context of these major environmental changes [68(p 98)].

During the Big Swamp phase the amount of *Pandanus spiralis*, an important food plant, within Madjebebe dramatically decreased [69 Fig 2]. This decrease is associated with poor preservation of organics more generally within this part of the stratigraphy so it is impossible to determine whether this is a post burial taphonomic effect or reflects the relative unavailability of *Pandanus* in the region at this time due to widespread mangroves and/or salt flats.

*Sinuous phase (4 to 2 ka).* During the Sinuous phase as mangrove forests migrated seaward away from the escarpment region there is a reciprocal reduction in mollusc resource exploitation observed in middens in this area. The decline in shell foraging does not occur simultaneously or at the same rate at all sites along the East Alligator River [1,15,16(p 102), 17(pp 75,110–45), 20,21,23(p 108), 24-26(pp 29–36,64–6)]. The rate of decline appears to be controlled by the sedimentation rates of the estuarine channels close to each site [25].

This phase shows a peak of the new lithic technology production through many of the sites in the region including Nawanmoyn, Madjedbebe and Ngarradj Warde Djobkeng. This change in technology supports the argument that these tools are a risk management strategy in the context of the highly variable environments and associated resources of this time [16,17,68].

There is evidence of site abandonment or long occupation hiatus at many of the sandstone rockshelter sites in the areas around the upper reaches of the East Alligator River estuary [16 (pp 90–1), 68(pp 96–7), 71]. This has been argued to represent a population shift towards the more open occupation sites closer to the coast following mangrove loss [25(p 212)]. This response may have been driven by the development of salt flats over areas where mangroves were receding [16(pp 90–1), 68(p 96),71(p 8)]. The timing of this site abandonment was synchronous with local mangrove decline. The timing of site abandonment/hiatus was different for each of the upper East Alligator River estuary sites with different abandonment times and rates observed for Madjedbebe and Ngarradj Warde Djobeng, likely reflecting mangrove decline patterns local to each site [25]. Reoccupation of the shelters local to the upstream sections of the East Alligator Estuary may not have occurred again until 1 ka [16(pp 90–1), 68(pp 96–7), 71]. Birriwulk's sustained occupation first occurred during this time.

*Cuspate phase (2 ka to present).* The Cuspate phase represents the most recent landscape usage, being the exploitation and occupation of the freshwater floodplains, which replaced the mangroves. A number of sites have evidence suggesting an increase in occupation during this time [13,22,24]. Ingaanjalwurr and MN005 also show evidence of occupation beginning during this phase. At MN05, which is located directly adjacent to Red Lily Lagoon, Wesley et al. [13(p 36)] reached sandstone bedrock at a depth of 95 cm with basal sediments dating to 795–950 Cal BP. This shows an accumulation of 1 m of sandy sediment over around 1000 years [13 (p 36)]. MN05 demonstrates a focus on freshwater species within the faunal remains with catfish (*Arius leptaspis)*, Barramundi (*Lates calcarifer)*, freshwater turtle, and freshwater bivalve (*Mytilus sp.)* within the excavated assemblage [13]. Biriwulk showed the same focus on freshwater resources with the presences of catfish and freshwater turtle within excavation layers dated between 750 and 50 BP.

Analysis of grinding stones dating to ~690 cal BP from Madjebebe demonstrates that starch from *Cochlospermum fraseri* was being processed at the site [70]. This species is usually found in open eucalypt woodland.

**Rock art.**   There are a great number of painted rock surfaces throughout the East Alligator River area. The rock art has significant variation in the styles and depicts a diversity of subject matter [2]. Rock art chronologies for Arnhem Land have been developed largely from the superimposition of relative sequences of motifs (younger paintings superimposed over the top of older paintings) [2]. Direct dating has confirmed that some anthropomorphic styles are Pleistocene and others emerged in the Pleistocene-Holocene transition [12]. The overall rock art chronology, however, is aligned with the broader environmental changes within the landscape. This is evident through changes in subject matter such as large naturalistic macropods to the appearance of estuarine animal species such as fish and crocodiles [2]. A proliferation of freshwater species such as fish and birds occur in the most recent styles of this region, and this reflects the most recent phases of environmental change with the extensive freshwater floodplains of the contemporary landscape [2].

**Geophysical investigation.** ERT provides a geophysical method of modelling the subsurface using induced electrical current [72, 73 (p 167)]. Current is induced and measured through electrodes partially inserted into the ground surface at a shallow depth (around 10-15cm). The arrangement of the current (injecting) and potential (measuring) electrodes is referred to as an array, and different array configurations will provide different samples of the subsurface resistivity. Typical array configurations include Wenner, Schlumberger and Dipole-Dipole [74]. Wenner is a robust array that is less sensitive to noise from electromagnetic disturbance and subsurface heterogeneity [32,33(pp 208–12)]. The Wenner array has a lower lateral resolution and will blur feature boundaries in this direction. The Schlumberger array is more sensitive to lateral boundaries and less sensitive to vertical changes [33(pp 208–12)]. The Dipole-dipole array is capable of deeper sounds from the same surface electrode coverage (per line measured). The Dipole-Dipole array is more sensitive to smaller features but also more sensitive to noise and disturbance from surface resistance heterogeneity [33(pp 208–12)].

Visualising the continuous distribution of subsurface resistivity values of a profile which can be interpreted in a meaningful way requires contouring of the data. This is the process of selecting discrete ranges of resistivity within the overall distribution and displaying these with a single unified colour [32]. The choice of resistivity values to be assigned to each group represented by a colour is important as one subsurface feature (such as a layer of sandy clay) may have a range of resistivity at different locations produced by changes in lithology, mineralogy, salinity and moisture [73,75–79]. Effective geophysical interpretation of ERT inversion requires choosing resistivity display brackets that appropriately breaks colours into distinct groups that reflect the subsurface characteristics of interest and are not separated by changes that are beyond the resolution of our interest (such as local changes in moisture content or salinity within a single sedimentary layer).

ERT can characterise subsurface stratigraphy of fluvial systems including different depositional facies and the sediment-bedrock interface [72, 73(p 167)]. However, the facies identified are based on resistivity of the features and so aren't informed by properties such as sedimentary structures or grain size, which can help inform stratigraphic interpretations using alternative methodologies (such as directly interpreting core samples) [78,79]. Thin or laterally confined stratigraphic units may also be below the resolution of ERT investigations [79].

ERT has great merit for the use in the East Alligator River floodplains. Following the geomorphological sequences that have been characterised for the floodplains of the South Alligator River and Magela Creek, the major depositional units that may be identified by this approach are bedrock (Kombolgie Sandstone or Koolpinyah Laterite), basal sands and gravels, fluvial sandy sediments (pre-transgression land surface), estuarine sandy clay (mangrove muds from the Big Swamp phase), and dominantly organic sediments (deposition from the Freshwater Floodplain environment) [41,42].

## Methods

### Ethics

This research was approved by Northern Land Council permit #79130 and via Flinders Human Research Ethics application #7704.

### Landscape modelling

The mapping of the contemporary land surface for the GRLLA study area was conducted using a Global Navigation Satellite System (GNSS) survey, drone-based photogrammetry and publicly available Light Detection and Ranging (LiDAR) data. Two digital elevation models

(DEM) were used. The largest DEM was derived from LiDAR collected at 1 m resolution and resampled to 5 m resolution (Geoscience Australia 2018) that has a vertical accuracy of at least 0.30 m and horizontal accuracy of at least 0.80 m (Geoscience Australia 2018). The smallest DEM was derived from drone-base photogrammetry. A DJI Mavic 2 Pro drone was used to undertake aerial photography. An Emlid RS+ RTK GNSS link and a CHC X90+ static GNSS were used to georeference the data. Structure from motion photogrammetry method was used to create the resulting DEM and orthophoto, which were processed using Agisoft Metashape software. The photogrammetry derived DEM had a spatial resolution of 26.4 cm This method is fully described in [80].

## ERT methods

**Geophysical field survey.** A geophysical investigation was conducted within the GRLLA study area in September 2019. September was chosen as this is the dry season of Northern Australia's yearly wet/dry tropical cycle and so has the least standing water across the floodplains. Four transects were chosen to sample the key landscape features that dominate the interface between the Kombolgie Sandstone escarpment and the directly adjacent East Alligator River floodplain. The features identified for geophysical survey are the floodplains, the incised valleys of the Kombolgie sandstone formation which run perpendicular to the floodplain, and the upland reaches of the sandstone valleys (See Fig 2 for examples of these features). Fig 4 shows the locations of the survey lines and Table 3 describes each line and their environmental and archaeological context.

The position of each electrode was recorded using an Emlid RS+ RTK GPS. The reported line lengths have demonstrated that minor cumulative errors in the 5 m spacing have caused less than 3 m of placement discrepancy per line. Electrodes were hammered approximately 10–20 cm into the ground and then connected together with cable. A ZZ FlashRES-Universal resistivity instrument was used to conduct the ERT survey. The contact resistance of each electrode was measured prior to survey. When this contact resistance was too high for a given electrode, saline solution was poured onto the ground around the base of the electrode until sufficient contact resistance was achieved. In general, contact resistances of <500 ohm were achieved on the floodplain. The ERT instrument was used to inject current and measure potential through the connected electrodes. This injection sequence was conducted in two distinct array patterns for each line, which were Wenner and Dipole-Dipole arrays. Each line was placed in a series of individual rolls of 64 electrodes spaced 5 m apart, after each roll the resistivity meter was connected to the next set of electrodes along the line with 32 electrodes overlapping each set of measurements.

**Data processing.** The data collected during the ERT field survey was filtered to remove values with extremely high or low resistivity. The threshold for judging outliers was considered on a line by line basis. Inversions were undertaken using Res2D software and used the L1-norm (robust) for the model. The surface elevation in metres above sea level according to the Australian Height Datum (AHD) was provided for each electrode and included in the inversion file to allow the resistivity profiles to be topographically corrected.

As each individual ERT survey line's inversion result has a unique distribution of resistivity, choosing a display colour range must be done with the full survey in mind so as to allow meaningful comparison between the different lines. To achieve this, all individual resistivity estimations across all survey lines were combined into a single dataset and a colour scale representing resistivity from $<1.35\ \Omega\cdot m$ to $>2771\ \Omega\cdot m$ was calculated using a quantile approach. This approach divided the data into 16 classes distributing observations equally across set intervals. This produces a set of classes with unequal widths but an equal frequency

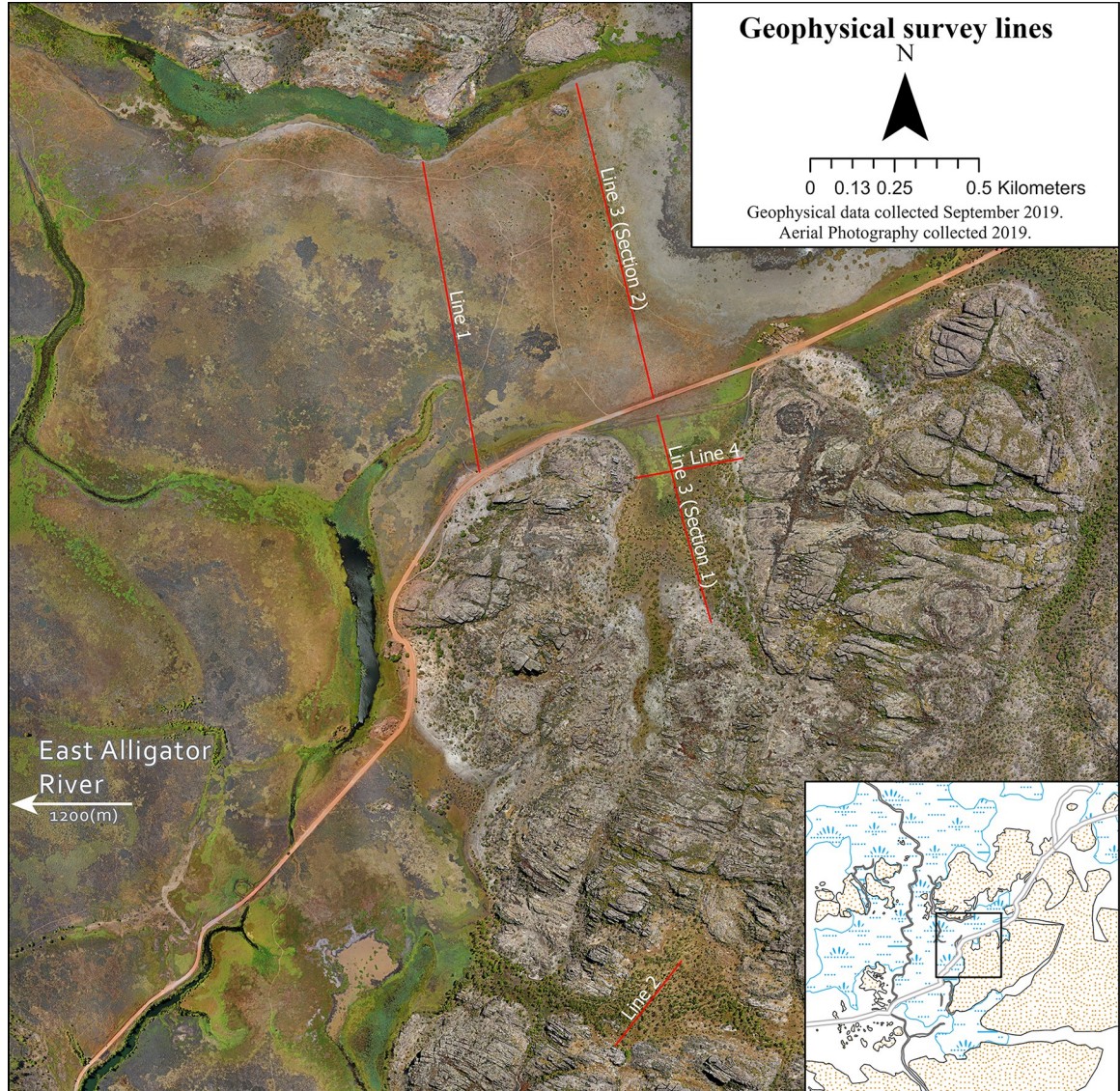

**Fig 4. Geophysical survey lines in the greater Red Lily Lagoon Area on the southeast edge of the East Alligator River floodplain.**

of observations per class. Each individual inversion result was then displayed using a unique colour for each of the 16 calculated classes. This allows comparison between different lines as values within the same colour contour are likely to belong to the same subsurface feature class.

## East Alligator River channel morphology

A LiDAR-derived DEM [81] was used to map both the modern channel configuration as well as palaeochannels with remnant visible surface depressions across an extended area of the East Alligator region. To maximise visibility of palaeochannels, the DEM was displayed with a stretched histogram using the 'histogram equalise' method. This stretch method was chosen as it accentuates local contrast without affecting the global contrast of the image. Palaeochannels were interpreted through inspection of this DEM and satellite imagery available through ESRI ArcGIS pro, and hand drawn.

**Table 3. ERT line locations.**

| Line # | Length | Description | Environment and archaeology |
|---|---|---|---|
| 1 | 982 m | Crosses the width of the floodplain from the sandstone escarpment. | Adjacent to a number of sites with extensive rock art. Crosses over the visible palaeochannel of Red Lily Lagoon's remnant waterway. |
| 2 | 314.52 m | Middle section of a valley through the stone plateau. | Forested with Melaleuca Paperbark trees.<br>This valley is a major drainage path for this section of the escarpment. An extensive number of rock art sites overlook this valley. |
| 3 (Section1) | 627.26 m | Open section of a major valley that runs into the stone plateau from the floodplains. | Melaleuca Paperbark open woodland.<br>The valley has a gentle slope descending towards the floodplain. Major rock art galleries surround this valley on all sides. |
| 3 (Section 2) | 953.67 m | Crosses the width of the floodplain from the escarpment valley to the northern escarpment edge. | Overlooked by several major rock art sites including a gallery with dated rock art panels [12]. |
| 4 | 312.21 m | Crosses the width of the sandstone valley. Crossing Line 3 (Section 1) at a perpendicular angle. | Melaleuca Paperbark open woodland.<br>The valley has a gentle slope descending towards the floodplain.<br>Major rock art galleries surround this valley on all sides. |

## Results and geophysical interpretation

**ERT results.** Fig 5 shows an ERT profile of survey lines 1, 3 and 4 displayed with consistent resistivity colour contouring for comparison. The profiles are plotted against elevation and therefore the surface topography of each line is also visible in the profiles.

**ERT interpretation.** The ERT profiles in Fig 5 show the distribution of resistivity in the subsurface for lines 1, 3 and 4. The inversion models were analysed in direct comparison to the expected depositional layers as derived from the floodplains of the South Alligator River and Magela Creek. Of the expected subsurface materials for the GRLLA floodplain environment, sandstone is likely to represent the feature with the highest resistivity closely followed by unconsolidated sand. Estuarine mud and organic sediments are likely to be less resistive. The interpretation of the underlying geomorphology for lines 1 and 3 can be seen in Fig 6.

**ERT floodplain interpretation.** Line 1 shows two extremely resistive features at the beginning and end of the line. These features have resistivity over 597 Ω·m. Based on their locations and shapes; these are interpreted as buried sandstone escarpments. The shape of these two features follows the forms of the exposed sandstone escarpments with their large sharply dropping terrace walls. This region of resistivity has been interpreted as Kombolgie sandstone, resistivity facies 'a' in Fig 6.

Adjacent to these sandstone features is a discrete area of moderate resistivity with a range of 197–597 Ω·m. These follow the form of the colluvial aprons of eroded sands from the Kombolgie sandstone which form against the escarpments. These regions of resistivity have been interpreted as colluvium, resistivity facies 'i' in Fig 6.

At the lowest levels of the profile is a region of distinct resistivity between 9 and 19 Ω·m. This feature has a surface elevation of around -4 m and has a geometry which correlates well to multiple incised fluvial channels. The surface height of this feature is within the elevation range of the pre-transgressive fluvial sediments characterised in Magela Creek (see Table 1 and Fig 3). This surface was described as sediments deposited in a freshwater fluvial system [41(p 29)]. The three large channels in this surface agree with this interpretation. The deepest point of the deepest channel in this feature is around -32 m elevation. These regions of resistivity have been interpreted as fluvial sediments, resistivity facies 'c', 'd', and 'e' in Fig 6.

Adjacent to and overlying these sediments is a highly resistive feature that spans from 140 m along the line until 575 m and continues to depths of ~40 m. It has resistivity between 28 Ω·m and 597 Ω·m. This overlaps the previously interpreted colluvial sands. Interestingly, this feature is directly in line with the palaeochannel which contains the modern Red Lily Lagoon.

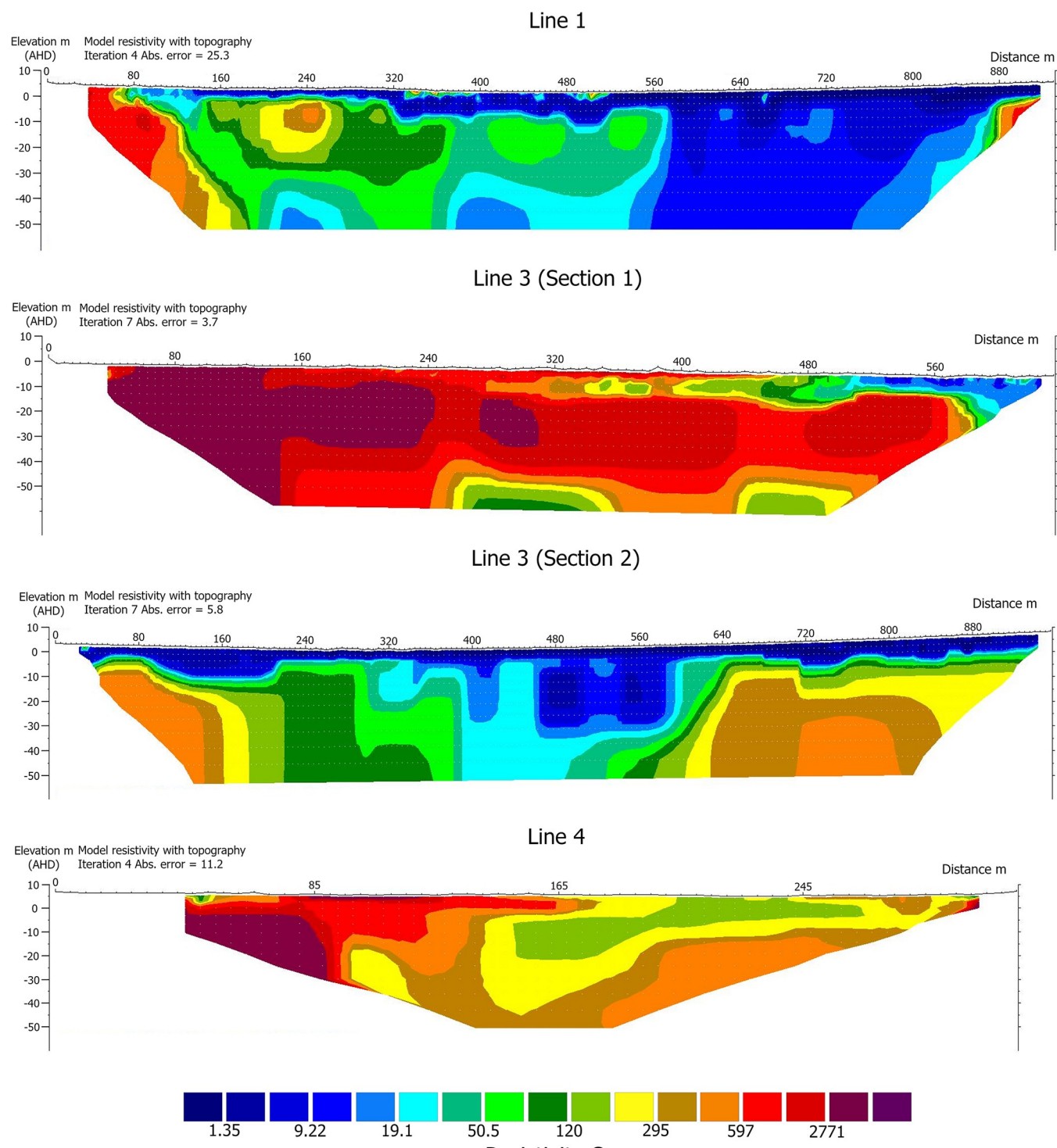

**Fig 5. ERT profiles from all survey lines collected using the Wenner array.** Resistivity has been contoured with colours grouping highly resistive features in dark red (>2771 Ω·m) and most conductive features in dark blue (<1.35 Ω·m). Areas of similar resistivity are shown in similar or the same colours presenting a map of resistivity facies in the subsurface.

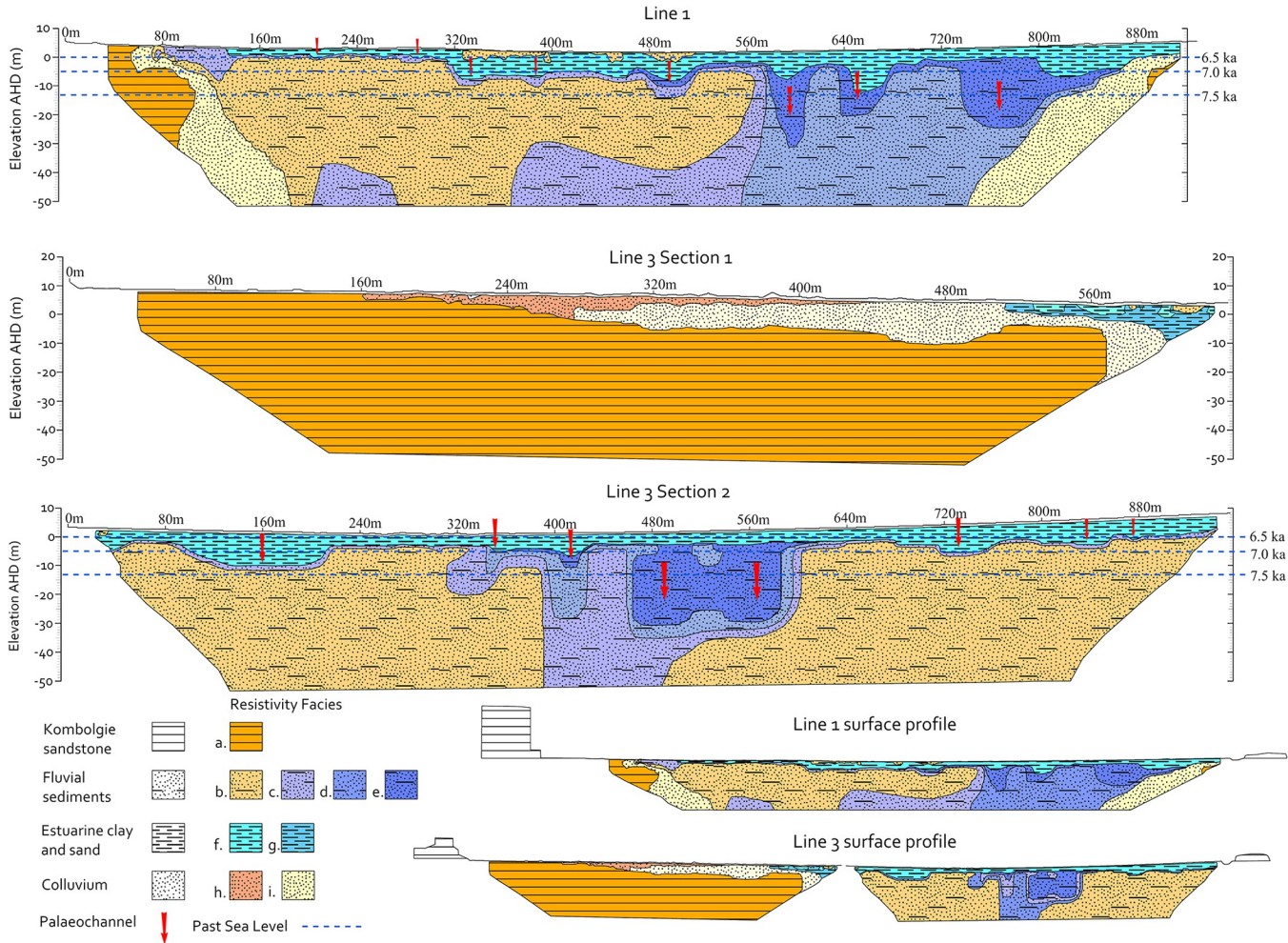

**Fig 6. Geophysical interpretation displaying subsurface units, resistivity facies and surface profiles.**

There are a number of channels visible in the top of this resistivity facies unit. The resistivity range and the presence of surface channels along with the location under the palaeochannel of Red Lily Lagoon suggest that this feature is alluvial sand-rich sediments deposited by aggradation during periods of fluvial activity. This feature is interpreted as fluvial sediments, resistivity facies 'b' in Fig 6.

The deepest channels in facies 'c', 'd' and 'e' are at a depth that aligns with the sandy fluvial sediments associated with the Red Lily Lagoon palaeochannel. This suggests that this palaeochannel may have been the principal course of the East Alligator River in the past. The contrast of resistivity between these two channel beds suggests a difference in grain size between these channels. This difference in sandy sediment may indicate that Red Lily Lagoon was the principal channel of the East Alligator River during this time, carrying the major sediment load from the larger Arnhem Plateau.

The buried fluvial sediments are overlain by a layer of more conductive material with a resistivity range of 0.2 Ω·m (the lowest recorded) to 9 Ω·m. This layer occurs between elevations of -13 and 5 m and infills channel-shaped features. The depths and the decreased resistivity suggest this layer is made up of estuarine clays and muds produced by the Transgressive phase and subsequent mangrove development during the Big Swamp phase. The elevation

range of this layer falls within the range of the mangrove sediments observed across the Magela Creek floodplain (see Fig 3) [41(pp 28–32,62)]. This layer has been interpreted as estuarine clay and sand, resistivity facies 'f', 'g' in Fig 6. These clays and sands occur adjacent to parts of the channel sediments of the Red Lily Lagoon palaeochannel. This suggests that while this channel predates the transgression, it continued acting as a distributary channel during the major mangrove infilling phase.

The transition from the Big Swamp to the contemporary freshwater floodplains has limited resistivity evidence. This absence may be caused by the resistivity of the materials being too similar to differentiate or the layer being too thin to have been modelled by the ERT inversion. There are several smaller pockets of higher resistivity materials (between 50 and 597 Ω·m) that occur sporadically along the top level of the resistivity profiles of the floodplain areas. These may represent evidence of the freshwater phase as they are in at the level of the contemporary surface and to a maximum depth of ~4 m. These features may represent palaeochannels from activity during the Freshwater phase with higher resistivity reflecting channel sediment deposits.

The interpretation of Line 1 can be compared directly to Line 3 (Section 2), which runs parallel to Line 1 and crosses the floodplains around 500 m east of Line 1. The same resistivity contours correlate to directly comparable subsurface features.

Line 3 (Section 2), which crosses the floodplain, shows two distinct regions of resistivity values that fall between 50 and 597 Ω·m (represented with green, yellow and orange colours in Fig 5) which are interpreted as sand-rich fluvial sediments. One of these two regions runs directly below Red Lily Lagoon and correlates to a similar feature in line 3 section 2 (seen in Fig 5 between 160 and 320 m along line 1 and 30 and 320 m along line 3 [Section 2]). The other sandy region does not correlate with Line 1. Inspection of the surface topography shows that there is a large, buried, meandering channel that hugs the sandstone escarpment on the northern edge of the floodplain. This is likely to be a continuation of the abandoned river channel in which Red Lily Lagoon is situated. The palaeochannel in which Red Lily lagoon is formed is therefore interpreted to have been the principal channel of the East Alligator River and acted as a distributary channel during the Big Swamp phase.

Between these two areas of higher resistivity is a lower resistivity feature with the geometry of two incised river channels. These two channels correlate with those in Line 1 that occur in the same band of lower resistivity fluvial sediments (facies 'e' incising through resistivity facies 'c' and 'd' in Fig 6). These channels are interpreted to have been incised into the pre-transgressive surface before being infilled with the much more conductive estuarine clays and mud of the Big Swamp phase. This feature is similar to the fluvial terrace structures of the pre-transgressive sediments observed in the Magela Creek floodplain [41(p 29)].

The estuarine sediments in Line 3 occur dominantly at elevations between -10 m and the surface level ~3.4 m. The small regions of higher resistivity which are interpreted as palaeochannels associated with the freshwater floodplain system are present in Line 3 at depths close to the surface but are less common than in Line 1.

The second half of Line 4 is dominated by resistivity values that fall between 50 and 597 Ω·m which are interpreted as sand-rich fluvial sediments. This may represent a meander loop of the fluvial system encroaching into the mouth of this valley.

**ERT valley slope interpretation.** Line 3 (Section 1; Fig 5) descends the gradual slope of a large valley in the sandstone. This ERT line has been displayed with the same colour scheme as the other lines and is based on the combined resistivity range. This line is dominated by a geographically extensive, highly resistive feature with resistivity between 597 and 2771 Ω·m (displayed in red in Fig 5). This is interpreted to represent a shallow, shallowly dipping sandstone shelf that underlies this valley at a depth of approximately 2–15 m before dropping vertically a

short distance into the floodplain. Some pockets of lower resistivity material occur within this sandstone shelf (represented in greens, yellows, and oranges at the bottom of Line 3 [Section 1] in Fig 5). These are likely to be part of the Kombolgie sandstone but represent localised variations in water saturation, porosity, or lithology. A band of highly resistive material, with resistivity between 597 and 1147 Ω·m, runs along the surface of the line in areas wooded with *Melaleuca* paperbark. Whilst these values are within the resistivity range interpreted as sandstone, observations of the sediment on the surface and the presence of mature trees suggest that this material is dry unconsolidated sand. The dry sandy sediment that fills the valley has been marked as colluvium, resistivity facies 'h' and 'i'. The transition between these facies ('h' and 'i' in Line 3 [Section 2]) is interpreted to represent local variations in water saturation and/or salinity.

At 480 m along Line 3 (Section 1), where this valley reaches the level of the floodplain, a band of conductive materials overlies the colluvium. These materials fall into the resistivity range interpreted as estuarine clay and muds of the Big Swamp phase. This band continues across the flood plain in Line 3 (Section 2). The sharp contrast between these resistivity facies shows that the colluvial apron against the sandstone escarpment was in place at the time of the transgression and was then overlain with the more rapid infilling of clays and muds. The shallow depth of the sandstone shelf in this valley suggests that some of the bedrock may have been exposed on the valley floor during the Pleistocene.

Line 2 is dominated by the same geographically extensive, highly resistive feature with resistivity between 597 and 2771 Ω·m seen in Lines 3 (Section 1) and 4. This is interpreted to represent a shallow, shallowly dipping sandstone shelf that underlies this valley at a depth of approximately 2–15 m.

The first half of Line 4 is dominated by the same highly resistive feature with resistivity between 597 and 2771 Ω·m seen in Lines 3 (Section 1) and 2. This is interpreted to represent a shallow, shallowly dipping sandstone shelf that underlies this entire valley immediately below the surface.

**Sea level rise and floodplain evolution.**   Mangroves in the modern Alligator rivers have been found to occupy an extremely restricted elevation range of 1 m below and 3.7 m above mean sea level [41,42,58,61]. A model of the sea level over time for this area was developed using the presence of mangrove materials in excavations of the South Alligator River floodplain [41(p 141), 58(p 124)]. Fig 6 shows some of the dated mean sea levels during the sea level rise following the LGM. On the basis of elevation, it is likely that in the GRLLA, the East Alligator River floodplain channels first became estuarine between 7.5 and 7 ka. By 6.5 ka, when the sea stabilised at its current level, sea water would have covered most of the modern floodplain especially during high tide. This led to the ubiquitous distribution of tidal flat sediments across the modern floodplains shown in the shallow subsurface by the ERT profile.

**East Alligator River channel morphology.**   Fig 7 shows a map of both the contemporary river channel and palaeochannels interpreted from the DEM and satellite imagery. Regions where former river meanders are visible on the surface have been labelled A through C. These regions occur where the bedrock topography allowed the formation of past river meanders and show a gradual progradation of the meandering section of the river morphology seaward.

## Discussion

### Pre-transgressive landscape

The ERT results present a detailed palaeogeography of the GRLLA over the 60 ky of Aboriginal occupation. The most striking feature in this landscape is a sandstone escarpment that stood above the level of the Pleistocene land surface that is now completely buried (See Line 1 in Fig

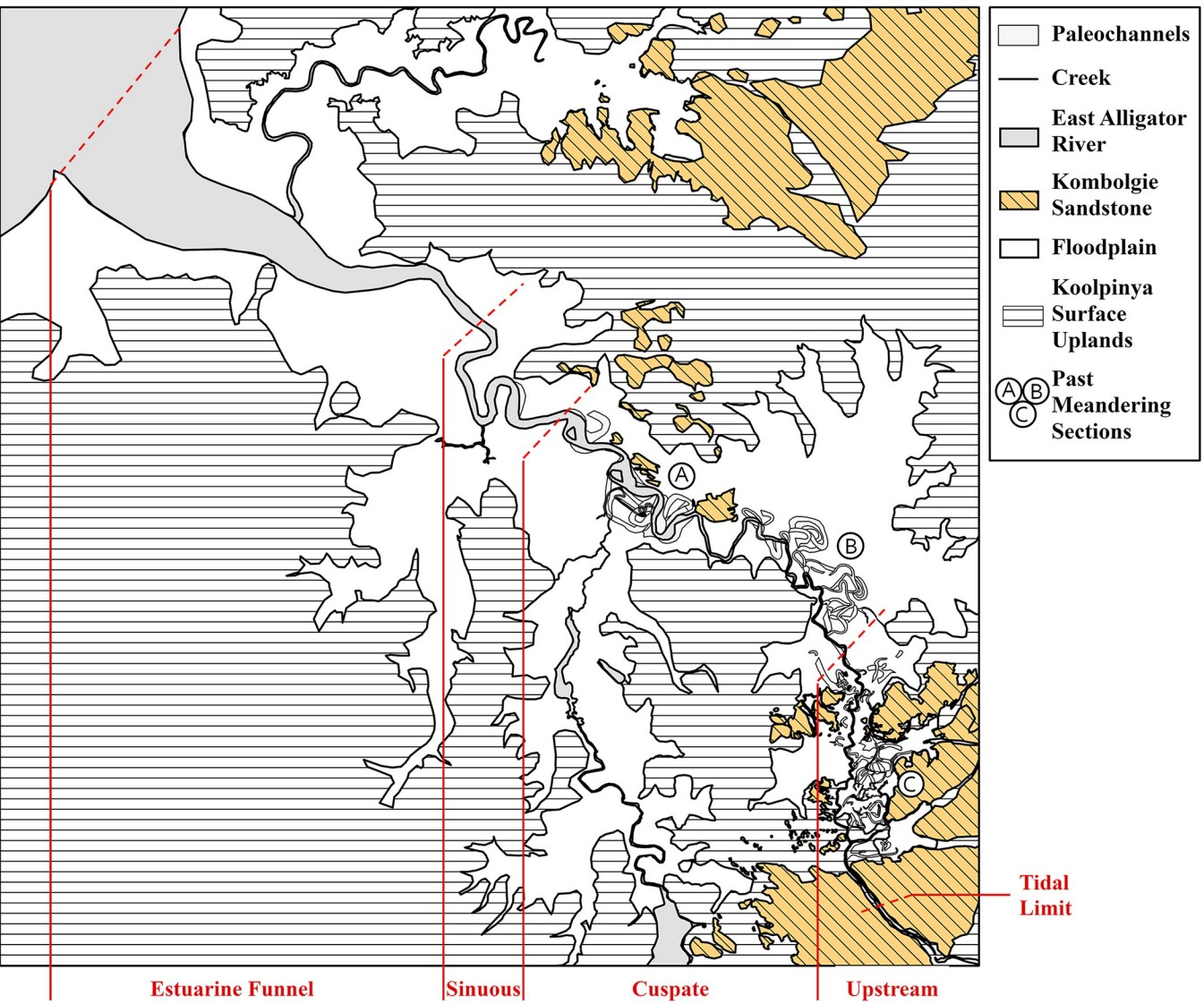

**Fig 7. East Alligator River channel and palaeochannel morphology.**

6). This escarpment was at a highly accessible location that was episodically adjacent to major river channels before the transgression and presents a likely location for human occupation during this time. If the subsurface geomorphology of the GRLLA is replicated elsewhere in the more than 280 km of floodplain-adjacent escarpment in Arnhem Land, it is likely that a great number of Pleistocene and Holocene sites of human habitation are now buried.

The results from Line 3 (Section 1) shows that this valley has a cover of unconsolidated sand over a bedrock surface that slopes towards the floodplain. Line 2 and the first half of Line 4 have very thin covers of unconsolidated sediment over a bedrock surface. The rate of the sediment accumulation in the valley that contains Lines 3 (Section 1) and Line 4 cannot be inferred through the ERT results; however, the stratigraphic relationship between this facies and the floodplain suggest that some of this material was deposited prior to the Big Swamp phase. This, in combination with the relatively thin accumulation of sediment, suggests a relatively slow sediment accumulation rate in this region. We have no means of determining the

age or accumulation rate of the unconsolidated sediment in the valley that contains Line 2 however the limited thickness of this unit suggests this region would not be prospective for buried stratified archaeological sites.

The ERT data show that sediment is being deposited in different ways (and at different rates) in different parts of the GRLLA, including the sheltered sandstone escarpments (MN05), sloping valleys within the larger sandstone formation (Line 3 [Section 1]), and the floodplains below the sandstone escarpments (Line 1 and Line 3 [Section 2]). This model is important in understanding the distribution of dates for occupation of shelter sites. All the excavated shelters were located on colluvium formed against the sandstone escarpments and elevated above the level of the transgressive infill and resulting sediments of the Big Swamp phase. These areas are good excavation targets on the bases of having slow rates of sediment infill and being close to the modern land surface, but they are not geographically extensive and only represent one aspect of the region's palaeogeography. This limited dataset has important implications for understanding settlement strategies over the human history of this region [16,68,71].

## Transgression and Big Swamp development

The ERT interpretation confirms that the Big Swamp mangroves extended to the outer reaches of the East Alligator River floodplain (and hence to the edge of the Arnhem Land escarpment) in the GRLLA. Some authors have suggested that this might be the case [41,58]. However, direct testing had not been conducted outside of areas close to the main channel of the South Alligator River and Magela Creek. This presence of Big Swamp sediments to the floodplain edges strongly supports the model of Woodroffe et al. [58(p 118)] of rapid transgressive infill even in distal parts of the floodplain. This rapid and extensive flooding would have had a significant impact on mobility and amount of land available for inhabitants of the GRLLA during the mid-Holocene.

The ERT profile demonstrates that the early period of the transgression was characterised by estuarine intrusion into the existing channels of the pre-transgressive surface. The rate of vertical aggradation of these channels must have been exceeded by the sea level rise by 6.5 ka. At this point sea water covered the entire floodplain, followed by mangrove development. This may provide a possible alternative mechanism for the presence of marine shellfish taxa found in Ngarradj Warde Djobkeng, Malangangerr and Nawamoyn, which have been interpreted as being traded from coastal origins as opposed to locally sourced [17, 25(pp 203,9,25)].

This extensive transgression event, and the subsequent decline of mangroves and establishment of salt flats in the region, must have had major implications for mobility and resource availability. This result strongly supports our contention that pandanus nut $\delta^{13}C$ [69] dated to this period reflect a response to increased standing water associated with the dramatic extents of marine influence and the effect on freshwater aquifer drainage.

## Sinuous and cuspate phases

The river morphology and palaeochannel mapping from the LiDAR data demonstrates that the configuration of the East Alligator River has changed significantly since its transition from a freshwater to a tidal-dominated estuary. The tidal-dominated estuary section has followed the overall straight-meandering-straight configuration. However, during the earliest periods of the Sinuous phase, the mixed-energy zone was clearly much closer to the stone plateau as demonstrated by the concentration of remnant river meanders in this region. This is consistent with the ERT interpretation that the influence of the transgression reached as far as the sandstone plateau in front of Red Lily Lagoon. Several meandering sections (Fig 7 sections a, b, c)

show the gradual movement of this mixed-energy zone downstream. These sections of river meander are likely to be separated by areas of bedrock control preventing meander development. This model of past river configuration goes further to demonstrate the major landscape changes over the human history of the region even in the past 5 ky.

The changes in the fluvial system during the Cuspate phase developed out of the erosion of past meandering sections and required the mixed-energy zone to have prograded before this development was possible. During this time the extensive salt flats that extended to the sandstone plateau are likely to have slowly transitioned to the seasonally inundated grasslands of the contemporary floodplain. However, there is no direct evidence of this development found within the data collected for this research.

## Approach assessment and limitations

The ERT results allow for a clear interpretation which strongly matches the existing models of the subsurface from South Alligator River and Magela Creek regions. The major limitation to the ERT was the inability to distinguish between the clays and organic sediments from the Big Swamp phase and the freshwater floodplain phase due to their similar resistivity character. Despite this, the depths of the estuarine clays and sands layers (facies 'f' and 'g') clearly match the depth ranges for the Big Swamp stratigraphy from the existing models.

## Implications for archaeological research in Western Arnhem Land

This research has several important implications for archaeological research in Western Arnhem Land. Firstly, we have demonstrated that the Pleistocene landscapes of this region can be effectively mapped using non-invasive methods. This has important implications for locating new sites but also for developing a more nuanced understanding of the regional palaeogeography, and its impact on human behaviour.

Based on the results of this study, all Pleistocene sites in western Arnhem Land on the edge of the escarpment were probably immediately adjacent to the ocean and, subsequently, mangrove swamps at some point during the transgression. This has important implications for the palaeogeographic settings of these sites, which must be considered when interpreting changes in stone artefacts, food resources and the isotope composition of biogenic materials from this period.

Further, the model of sandstone valleys being filled with a slowly accumulating apron of locally derived sand which we observe in ERT lines 2, 3 and 4 is important for understanding the archaeology of Western Arnhem Land more broadly. All archaeological sites with Pleistocene dates within the region (Madjedbebe, Ngarradj Warde Djobkeng, Malangangerr, Nawamoyn, Bindiarran and Birriwulk) are associated with similar colluvial aprons adjacent to sandstone escarpments at elevations higher than the floodplain. This is in keeping with our proposed model that this is the only position in this landscape that Pleistocene sediments will exist close to the modern land surface in the region.

Finally, our survey reveals the tantalizing prospect of a buried escarpment with adjacent thick accumulations of Pleistocene aged sediment. This geomorphic feature almost certainly contains abundant archaeological sites. However, its position beneath up to 9 m of floodplain sediment means that it probably remains out of reach of archaeological surveys with modern technology.

## Future directions

The timings of the mangrove development detected by ERT was inferred from the sea level curves developed from the published excavations in the South Alligator River and Magela

Creek regions [19,41,42]. However, direct investigation and dating of the landforms characterised for the pre-Transgressive phase and the subsequent freshwater floodplain development would help to characterise and validate the models developed through this research.

The 5 m electrode spacing of the survey allowed large cross sections of the floodplain to be covered; however, this came at the cost of resolution. This was suitable for the aims of this research to characterise floodplain stratigraphy in a broad sense. A more detailed ERT survey with shorter electrode spacing could allow the further characterisation of the morphology as well as further detection and characterisation of palaeochannels. The landscape profile also has the potential to show likely locations for middens. Middens may be detectable using ERT, further increasing the potential for ERT investigation of the floodplains for site detection [82].

## Conclusions

ERT is a rapid, low-cost, non-invasive method that can characterise large areas of the landscape. ERT data can be used to develop landscape models that are useful in understanding known site locations as well as predicting new site locations. Combined ERT- and LiDAR-based modelling characterised the major landscape changes from the late Pleistocene to the late Holocene of the Greater Red Lily Lagoon Area. This characterisation helped connect broader landscape models to the floodplain surrounding Red Lily Lagoon and extended previous models of the transgression and its impacts on previous land surfaces. Directly demonstrating the burial of a sandstone escarpment in close proximity to a major stream channel present during the late Pleistocene and early Holocene is an important step to modelling the landscapes and human settlement strategies. The characterisation of the colluvial valleys and aprons of the sandstone formations demonstrates a useful model in understanding the locations and taphonomy of the known older pre-transgressive occupation sites such as Madjedbebe.

## Acknowledgments

Thank you to the Njanjma Rangers who provided outstanding support for the research.

## Author Contributions

**Conceptualization:** Jarrad Kowlessar, Ian Moffat, Daryl Wesley, Alfred Nayinggul.

**Data curation:** Jarrad Kowlessar.

**Formal analysis:** Jarrad Kowlessar.

**Funding acquisition:** Ian Moffat, Daryl Wesley, Tristen Jones.

**Investigation:** Jarrad Kowlessar, Ian Moffat, Daryl Wesley, Mark Willis, Shay Wrigglesworth, Alfred Nayinggul.

**Methodology:** Jarrad Kowlessar, Ian Moffat, Daryl Wesley, Mark Willis, Shay Wrigglesworth, Alfred Nayinggul.

**Project administration:** Ian Moffat, Daryl Wesley, Alfred Nayinggul.

**Resources:** Ian Moffat, Daryl Wesley, Alfred Nayinggul.

**Supervision:** Ian Moffat, Daryl Wesley, Alfred Nayinggul.

**Validation:** Jarrad Kowlessar, Alfred Nayinggul.

**Visualization:** Jarrad Kowlessar, Mark Willis.

**Writing – original draft:** Jarrad Kowlessar.

**Writing – review & editing:** Jarrad Kowlessar, Ian Moffat, Daryl Wesley, Mark Willis, Shay
Wrigglesworth, Tristen Jones, Alfred Nayinggul.

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
