## [Decision Letter · Decision Letter 0]

3 Feb 2023

PONE-D-22-27065Reconstructing archaeological palaeolandscapes using geophysical and geomatic survey techniques: An example from Red Lily Lagoon, Arnhem Land, AustraliaPLOS ONE

Dear Dr. Moffat,

Thank you for submitting your manuscript to PLOS ONE. After careful consideration, we feel that it has merit but does not fully meet PLOS ONE’s publication criteria as it currently stands. Therefore, we invite you to submit a revised version of the manuscript that addresses the points raised during the review process.

 Two reviewers provided comments on your paper. As you will see, the recommendations for improvement are relatively minor. Please consider these points in your revision.  

We look forward to receiving your revised manuscript.

Kind regards,

Michael D. Petraglia, Ph.D.

Academic Editor

PLOS ONE

Journal Requirements:

"Jarrad Kowlessar is the recipient of a Flinders University Postgraduate Scholarship. Ian Moffat, Daryl Wesley and Tristen Jones were supported by George Chaloupka Fellowships from the Museum and Art Gallery of the Northern Territory. Daryl Wesley is the recipient of an Australian Research Council Discovery Early Career award (project number DE170101447) funded by the Australian Government. Ian Moffat is the recipient of an Australian Research Council Discovery Early Career award (project number DE160100703) funded by the Australian Government and a Flinders University Early Career Researchers Award. This research was approved by Northern Land Council permit #79130 and via Flinders Human Research Ethics application #7704. Thank you to the Njanjma Rangers who provided outstanding support for the research."

"JK is the recipient of a Flinders University Postgraduate Scholarship (flinders.edu.au).

IM is the recipient of an Australian Research Council Discovery Early Career award (project number DE160100703, arc.gov.au) funded by the Australian Government, a George Chaloupka Fellowship from the Museum and Art Gallery of the Northern Territory (magnet.net.au) and a Flinders University Early Career Researchers Award (flinders.edu.au).

DW is the recipient of an Australian Research Council Discovery Early Career award (project number DE170101447, arc.gov.au) funded by the Australian Government and a George Chaloupka Fellowship from the Museum and Art Gallery of the Northern Territory (magnet.net.au).

TJ is the recipient of a George Chaloupka Fellowship from the Museum and Art Gallery of the Northern Territory (magnet.net.au).

4. We note that Figures 1, 4 and 7 in your submission contain [map/satellite] images which may be copyrighted. All PLOS content is published under the Creative Commons Attribution License (CC BY 4.0), which means that the manuscript, images, and Supporting Information files will be freely available online, and any third party is permitted to access, download, copy, distribute, and use these materials in any way, even commercially, with proper attribution. For these reasons, we cannot publish previously copyrighted maps or satellite images created using proprietary data, such as Google software (Google Maps, Street View, and Earth). For more information, see our copyright guidelines: http://journals.plos.org/plosone/s/licenses-and-copyright.

   1. You may seek permission from the original copyright holder of Figure 1, 4 and 7 to publish the content specifically under the CC BY 4.0 license. 

Reviewers' comments:

Reviewer's Responses to Questions

**Comments to the Author**

1. Is the manuscript technically sound, and do the data support the conclusions?

Reviewer #1: Yes

Reviewer #2: Yes

2. Has the statistical analysis been performed appropriately and rigorously? 

Reviewer #1: Yes

Reviewer #2: Yes

3. Have the authors made all data underlying the findings in their manuscript fully available?

Reviewer #1: Yes

Reviewer #2: Yes

4. Is the manuscript presented in an intelligible fashion and written in standard English?

Reviewer #1: Yes

Reviewer #2: Yes

5. Review Comments to the Author

Reviewer #1: This is a very important paper and a first of its kind, using ERT to map the sub-surface geomorphic units and phases of the transgression in the GRLLA. The findings that all major Pleistocene sites were likely immediately adjacent to the ocean and mangrove forests at some point during the transgression provides a vital clue in interpreting changing landuse, technology and occupational intensity in the region over the last 60,000 years. This approach could be more widely adopted in the region, and opens the possibility of future excavations close to the escarpment edge adjacent to Red Lilley Lagoon with high probability of encountering buried Pleistocene sites (and rockshelters?). The non-invasive nature of this approach is likely to assure its success in future research in the region. This paper should be published with minor revisions. I have a number of small comments and suggested edits below.

Pge 3, line 72, reference to occupation by 60ka, but later 65ka. SHold be 65ka here.

LInes 79,80 Finding new sites will probably enable testing of the stratigraphic integrity of Madjedbebe per se, only replicating its sequence or not.

Line 102, MSI should be MIS?

Table 2, Madjedbebe max occupation given as 61+/- 10ka. Modelled start and end ages in Clarkson et al. 2017 are given as 65.0 ± (3.7, 5.7) and 52.7 ± (2.4, 4.3) kyr.

Line 389 New grindstone paper and new archaeobotany papers could also be cited here.

Florin, S.A., Fairbairn, A.S., Nango, M., Djandjomerr, D., Hua, Q., Marwick, B., Reutens, D.C., Fullagar, R., Smith, M., Wallis, L.A. and Clarkson*, C., 2022. 65,000 years of changing plant food and landscape use at Madjedbebe, Mirarr country, northern Australia. Quaternary Science Reviews, 284, p.107498.

Hayes, E.H., Fullagar, R., Field, J.H., Coster, A.C., Matheson, C., Nango, M., Djandjomerr, D., Marwick, B., Wallis, L.A., Smith, M.A. and Clarkson*, C., 2022. 65,000-years of continuous grinding stone use at Madjedbebe, Northern Australia. Scientific Reports, 12(1), pp.1-17.

Line 462 (and 824-828). Although it is possible pandanus declines in use or abundance in the landscape at this time, the results in Florin cannot be interpreted this way. The reason abundance dropped out is that charcoal does not survive well in Phase 6, and hence there is almost no surviving analysable carbonised archaeobotanics from this period. See Florin 2020.

Table 3 rockart inscription. is that the right word?

Line 624 extend should be extended

Reviewer #2: General comments

This paper deals with palaeogeographical reconstructions and the development of methodological approaches to better understand palaeolandscape evolution and its impact on the early human settlement of Australia. Changing landscapes have proved to affect greatly exploitation, subsistence and dispersal of hominin and early human populations, yet this parameter only recently has started to be factored in in readings and interpretations of the archaeological evidence from the Palaeolithic period. The methodological challenges are usually great and deep time reconstructions are difficult (e.g. lack of high resolution datasets – environmental, geological or fragmented archaeological records etc.) and so the development of new approaches is helping this wider discussion to move forward.

ERT as a method is not new and the authors are very well aware of the methodology’s potentials and limitations (lines 535-559). Yet the paper demonstrates the effectiveness of the ERT method – that emerges as a significant tool: (a) to investigate palaeolandscapes and their evolution over time (both short and long-term changes); (b) to predict new areas of Pleistocene/pre-Holocene activity; (c) to understand better human settlement strategies; (d) to offer new/alternative interpretations for the archaeological record in relation to rapid changes over the palaeolandscape (e.g. transgression episode 6,5 Ka and the presence of marine sell fish or observed hiatus in settlement associated with the decrease of habitable land due to the establishment of salt flats). Ultimately, this work allows predictions of new ‘targets’ for the future archaeological research that need to be confirmed or rejected through fieldwork.

The methodology detailed in the paper can be applied in other similar early contexts beyond Australia, to answer similar research questions. The paper promotes interdisciplinarity using earth sciences methodologies to offer new insights in behavioural parameters relevant to settlement choices.

This is a significant work that definitely deserves to be accepted for publication - with some minor revisions.

Suggestions for minor revisions

- Introduction> I would suggest considering minor restructuring. My suggestion would be to promote the archaeological info and then discuss methodologies and their contribution in reconstructing the palaeolandscape. It needs to be clear, I believe, why the palaeolandscape under study is important and why it would be useful to reconstruct it and try to locate pre-Holocene sites. Perhaps consider implementing lines 35-48 into the last paragraph from line 92 onwards.

- A general map of Australia with the study area shown/highlighted would be extremely useful for international readers. Perhaps consider adding it.

- Some minor issues with dates/dating methods:

Lines 279-280 > how do these dates occur, please mention the dating method.

Lines 297-299 > the basal sands and gravel layer is consisted of pre-Holocene sediments, are there any more details on the chronology/dating of these sediments? However, in Table 1 the chronology of the same layer is noted as ‘unknown’ – I would have expected pre-Holocene (?), if my understanding is correct. In any case, this needs to be clearer.

Lines 338-340 > dating method of sediments?

Line 393 > Ka is missing after 35

- Line 535-6> repetition of ‘in a meaningful way’ consider rephrasing for clarity

- Lines 617-18> perhaps elaborate a bit more on inversion result (in one-two sentences). How do the classes occur? It’s important to be clear and for the reader to understand it.

- Lines 635-6 > is line 2 profile missing from fig 5?

6. PLOS authors have the option to publish the peer review history of their article (what does this mean?). If published, this will include your full peer review and any attached files.

Reviewer #1: No

Reviewer #2: **Yes: **Peny Tsakanikou

---

## [Author Response · Author response to Decision Letter 0]

28 Feb 2023

We have amended the manuscript to ensure it conforms to Plos One style requirements.

2. Please remove any funding-related text from the manuscript and let us know how you would like to update your Funding Statement. Currently, your Funding Statement reads as follows.

We have removed all funding-related text from the manuscript but don’t require any changes to the existing funding statement.

3. Provide the data repository details

DOI: https://doi.org/10.17605/OSF.IO/8PJER

4. Please provide permissions for Figures 1, 4 and 7

Figures 4 and 7 are based entirely on our own data and so do not require permissions. Figure 1 has been modified so it now contains only our own data and so does not require permissions.

5. Review the reference list

We have added a number of additional papers to the reference list.

6. Pge 3, line 72, reference to occupation by 60ka, but later 65ka. SHold be 65ka here.

Agreed, we have amended.

7. LInes 79,80 Finding new sites will probably enable testing of the stratigraphic integrity of Madjedbebe per se, only replicating its sequence or not.

Agreed, we have amended the text to reflect this.

8. Line 102, MSI should be MIS?

Agreed, we have amended.

9. Table 2, Madjedbebe max occupation given as 61+/- 10ka. Modelled start and end ages in Clarkson et al. 2017 are given as 65.0 ± (3.7, 5.7) and 52.7 ± (2.4, 4.3) kyr.

Agreed, we have amended

10. Line 389 New grindstone paper and new archaeobotany papers could also be cited here.

Florin, S.A., Fairbairn, A.S., Nango, M., Djandjomerr, D., Hua, Q., Marwick, B., Reutens, D.C., Fullagar, R., Smith, M., Wallis, L.A. and Clarkson*, C., 2022. 65,000 years of changing plant food and landscape use at Madjedbebe, Mirarr country, northern Australia. Quaternary Science Reviews, 284, p.107498.

Hayes, E.H., Fullagar, R., Field, J.H., Coster, A.C., Matheson, C., Nango, M., Djandjomerr, D., Marwick, B., Wallis, L.A., Smith, M.A. and Clarkson*, C., 2022. 65,000-years of continuous grinding stone use at Madjedbebe, Northern Australia. Scientific Reports, 12(1), pp.1-17.

Agreed, we have added these as well as the (very) new Langley et al. 2023 paper.

11. Line 462 (and 824-828). Although it is possible pandanus declines in use or abundance in the landscape at this time, the results in Florin cannot be interpreted this way. The reason abundance dropped out is that charcoal does not survive well in Phase 6, and hence there is almost no surviving analysable carbonised archaeobotanics from this period. See Florin 2020.

Agreed, we have amended the text in both of these sections to reflect this.

12. Table 3 rockart inscription. is that the right word?

We agree this isn’t the right word, we have amended.

13. Line 624 extend should be extended

Agreed, we have amended.

14. Introduction> I would suggest considering minor restructuring. My suggestion would be to promote the archaeological info and then discuss methodologies and their contribution in reconstructing the palaeolandscape. It needs to be clear, I believe, why the palaeolandscape under study is important and why it would be useful to reconstruct it and try to locate pre-Holocene sites. Perhaps consider implementing lines 35-48 into the last paragraph from line 92 onwards.

Agreed, we have amended.

15. A general map of Australia with the study area shown/highlighted would be extremely useful for international readers. Perhaps consider adding it.

Agreed, we have replaced the inset map of the Northern Territory in Fig 1 to one of Australia.

16. Lines 279-280 > how do these dates occur, please mention the dating method.

We amended this text to include the dating method.

17. Lines 297-299 > the basal sands and gravel layer is consisted of pre-Holocene sediments, are there any more details on the chronology/dating of these sediments? However, in Table 1 the chronology of the same layer is noted as ‘unknown’ – I would have expected pre-Holocene (?), if my understanding is correct. In any case, this needs to be clearer.

Agreed, we have amended this text to make it clearer.

18. Lines 338-340 > dating method of sediments?

We amended this text to include the dating method

19. Line 393 > Ka is missing after 35

Agreed, we have added Ka

20. Line 535-6> repetition of ‘in a meaningful way’ consider rephrasing for clarity

Agreed, we have deleted one instance of this

21. Lines 617-18> perhaps elaborate a bit more on inversion result (in one-two sentences). How do the classes occur? It’s important to be clear and for the reader to understand it.

Agreed, we have added more text about this

22. Lines 635-6 > is line 2 profile missing from fig 5?

We haven’t displayed Line 2 because it replicates the bedrock shelf in Line 3 (Section 1) and Line 4. We have added more text to make this explicit.

---

## [Editor Report · Decision Letter 1]

1 Mar 2023

Reconstructing archaeological palaeolandscapes using geophysical and geomatic survey techniques: An example from Red Lily Lagoon, Arnhem Land, Australia

PONE-D-22-27065R1

Dear Dr. Moffat,

We’re pleased to inform you that your manuscript has been judged scientifically suitable for publication and will be formally accepted for publication once it meets all outstanding technical requirements.

Kind regards,

Michael D. Petraglia, Ph.D.

Academic Editor

PLOS ONE
---

## [Editor Report · Acceptance letter]

3 Apr 2023

PONE-D-22-27065R1 

Reconstructing archaeological palaeolandscapes using geophysical and geomatic survey techniques: An example from Red Lily Lagoon, Arnhem Land, Australia 

Dear Dr. Moffat:

I'm pleased to inform you that your manuscript has been deemed suitable for publication in PLOS ONE. Congratulations! Your manuscript is now with our production department. 

Kind regards, 

on behalf of

Professor Michael D. Petraglia 

Academic Editor

PLOS ONE